# *V. cholerae* MakA is a cholesterol-binding pore-forming toxin that induces non-canonical autophagy

Xiaotong Jia[1,2] , Anastasia Knyazeva[1,2], Yu Zhang[1,2], Sergio Castro-Gonzalez[1,2], Shuhei Nakamura[3], Lars-Anders Carlson[2,4,5,6], Tamotsu Yoshimori[3] , Dale P. Corkery[1,2] , and Yao-Wen Wu[1,2]

**Pore-forming toxins (PFTs) are important virulence factors produced by many pathogenic bacteria. Here, we show that the *Vibrio cholerae* toxin MakA is a novel cholesterol-binding PFT that induces non-canonical autophagy in a pH-dependent manner. MakA specifically binds to cholesterol on the membrane at pH < 7. Cholesterol-binding leads to oligomerization of MakA on the membrane and pore formation at pH 5.5. Unlike other cholesterol-dependent cytolysins (CDCs) which bind cholesterol through a conserved cholesterol-binding motif (Thr-Leu pair), MakA contains an Ile-Ile pair that is essential for MakA-cholesterol interaction. Following internalization, endosomal acidification triggers MakA pore-assembly followed by ESCRT-mediated membrane repair and V-ATPase-dependent unconventional LC3 lipidation on the damaged endolysosomal membranes. These findings characterize a new cholesterol-binding toxin that forms pores in a pH-dependent manner and reveals the molecular mechanism of host autophagy manipulation.**

## Introduction

*Vibrio cholerae*, a Gram-negative bacterium that inhabits aquatic environments, is the bacterial pathogen responsible for the diarrheal disease cholera (Faruque et al., 1998). The ability of *V. cholerae* strains to cause severe epidemics in humans requires the expression of cholera toxin (Ctx) and toxin co-regulated pilus (TCP; Kaper et al., 1995). Besides Ctx and TCP, *V. cholerae* also produces additional secreted toxins. Motility associated killing factor A (MakA) is a newly discovered cytotoxin from *V. cholerae* (Dongre et al., 2018). It was shown to function as a virulence factor pathogenic to *C. elegans* and zebrafish (Dongre et al., 2018), and co-expressed with four other Mak proteins in a *V. cholerae* gene cluster, *makDCBAE* (*vca0880–vca0884*; Nadeem et al., 2021).

Pore-forming toxins (PFTs) are the largest class of bacterial toxins and are important virulence factors produced by many pathogenic bacteria (Dal Peraro and van der Goot, 2016). A large group of PFTs have been described, all characterized by their ability to form pores within membranes after oligomerization. In most reported cases, these toxins are secreted by pathogens in a soluble, monomeric form that can bind to target cells using receptors (lipids/sugars/proteins), followed by their oligomerization into an amphipathic structure with an arc or pore shape

that inserts into membrane (Iacovache et al., 2008). Membrane insertion may trigger a series of responses in the host cell, including cellular ion imbalance, membrane damage/repair, autophagy, and so on (Bischofberger et al., 2012).

Autophagy is an evolutionarily conserved cellular process involving the delivery of cytoplasmic components, such as dysfunctional organelles or protein aggregates, to lysosomes for degradation. During autophagy, intracellular constituents are sequestered within double-membraned autophagosomes, which fuse with endosomal and lysosomal compartments, resulting in degradation of engulfed material by lysosomal hydrolases (Mizushima and Komatsu, 2011; Dikic and Elazar, 2018; Klionsky, 2021). Autophagosome biogenesis is regulated by a series of core autophagy machinery, including the initiation complex (ULK1/ATG13/FIP200/ATG101), the PI(3)Kinase complex (VPS34/VPS15/BECN1/ATG14L), and the E3-like conjugation complex (ATG12-ATG5-ATG16L1), which work in concert to catalyze the conjugation of microtubule-associated protein light chain 3 (LC3) to phosphatidylethanolamine (PE) on nascent autophagosomal membranes (Mizushima et al., 2011). While the covalent attachment of LC3 to PE on autophagosomal membranes remains the hallmark of macroautophagy, LC3 lipidation

.....................................................................................................................................................................................................................

[1]Department of Chemistry, Umeå University, Umeå, Sweden;   [2]Umeå Centre for Microbial Research, Umeå University, Umeå, Sweden;   [3]Department of Genetics, Graduate School of Medicine, Osaka University, Osaka, Japan;   [4]Department of Medical Biochemistry and Biophysics, Umeå University, Umeå, Sweden;   [5]Wallenberg Centre for Molecular Medicine, Umeå University, Umeå, Sweden;   [6]Molecular Infection Medicine Sweden, Umeå University, Umeå, Sweden.

Correspondence to Yao-Wen Wu: yaowen.wu@umu.se;   Dale P. Corkery: dale.corkery@umu.se.

has been shown to occur in response to activation of several non-degradative pathways (Galluzzi and Green, 2019). These "non-canonical" autophagy pathways occur independent of some of the core autophagy machinery and generally do not require formation of double-membraned autophagosomes.

Both canonical and non-canonical autophagy pathways play an important role in the host defense against microbial pathogens. These roles range from the direct elimination of intracellular pathogens trapped within a double-membraned autophagosome (Xenophagy; Randow and Youle, 2014) to the regulation of antigen presentation by LC3-associated phagocytosis (LAP; Heckmann and Green, 2019), or cytokine secretion by secretory autophagy pathways (Ponpuak et al., 2015). As a result, many bacterial pathogens have evolved sophisticated strategies to avoid or combat host autophagy (Baxt et al., 2013; Choy et al., 2012; Ogawa et al., 2005; Wu and Li, 2019; Yang et al., 2017).

Our recent study reported a novel mechanism of autophagy modulation by the *V. cholerae* cytotoxin, MakA (Corkery et al., 2021). We show that MakA is endocytosed into host cells resulting in the formation of cholesterol-rich endolysosomal membrane aggregates. Aggregate formation induces a non-canonical autophagy pathway leading to the unconventional LC3 lipidation on endolysosomal membranes.

In this work, we aimed to further investigate the molecular mechanism of MakA-induced non-canonical autophagy in cells. We found that MakA binds to cholesterol on the membrane in a pH-dependent manner and that low pH triggers MakA assembly into a pore within the membrane. A key motif (Ile-Ile pair) has been identified to be essential for MakA-cholesterol interaction. Loss of cholesterol binding or inhibition of endosomal acidification was shown to prevent MakA-induced pore formation and subsequent activation of non-canonical autophagy. These findings reveal that MakA is a novel cholesterol-binding toxin that can form pores in a pH-dependent manner.

## Results

### MakA binds to cholesterol on the membrane in a pH-dependent manner

As pH-dependent endocytosis of MakA alters the cellular cholesterol distribution and induces cholesterol-rich endolysosomal membrane aggregate formation in host cells (Corkery et al., 2021), we hypothesized that MakA might interact directly with cholesterol on the membrane. To confirm this hypothesis, we made use of liposome sedimentation assay with multilamellar vesicles to test the interaction between MakA and membrane lipids at different pHs. At pH 6.8, which is in the functional pH range of MakA-induced LC3 lipidation (Corkery et al., 2021), native MakA was precipitated after incubation with liposomes containing phosphatidylcholine (PC) and cholesterol. The binding efficiency was gradually enhanced when concentrations of cholesterol was increased from 10 to 70% (Fig. 1 A). In contrast, at pH 6.8, no obvious binding was observed between MakA and other lipids, such as phosphocholine (PC), phosphatidic acid (PA), sphingomyelin, phosphatidylethanolamine (PE), phosphatidylserine (PS), or phosphatidylinositol (PI; Fig. 1 B).

Interestingly, MakA's interaction with cholesterol on liposomes was found to be strongly dependent on pH. In liposomes containing 70% cholesterol, when pH was decreased from 7.6 to 5.4, the amount of precipitated MakA was dramatically increased (Fig. S1 A). In contrast, binding of MakA to liposomes with 100% PC was not observed at any pH (7.6–5.4; compared to the negative control in buffer, Fig. 1 C). Next, we analyzed the pH-dependent interaction between MakA and membrane using PM mix liposomes containing PC/PE/PS/PI/cholesterol/sphingomyelin (5:1:0.5:0.5:2:1), which emulate the lipid composition of the plasma membrane in mammalian cells. Similar to liposomes with 70% cholesterol, binding of MakA to PM mix liposomes took place at pH 7.0, significantly increased to around 80% at pH 6.8 and continued to reach more than 95% at pH 5.4 (Fig. 1 C). Similar results were obtained using the liposome flotation assay (Fig. S1 B). Furthermore, we observed pH-dependent binding of Alexa568-labeled MakA (Alexa568_MakA) to cholesterol on the giant unilamellar vesicles (GUVs). Alexa568-MakA showed co-localization with GUVs containing cholesterol at pH 5.4, whereas there was almost no MakA co-localization observed at pH 7.6 (Fig. 1 D). Consistent with our earlier findings (Corkery et al., 2021), we observed co-localization of Alexa568-MakA with cholesterol-rich membranes (filipin staining) in all cell lines tested (Fig. 1 E). Taken together, these data confirm that MakA binds to membrane cholesterol in a pH-dependent manner.

### Kinetics and affinity of MakA-cholesterol interaction

To investigate the dynamics of MakA's interaction with cholesterol on the membrane, quartz crystal microbalance with dissipation (QCM-D) analysis was performed to determine binding of MakA to a supported lipid bilayer (SLB) with the same composition of PM mix liposomes. After MakA injection, the saturated amount of MakA detected on SLB was increased from pH 6.8–5.4 (Fig. S2 A), which is consistent with results from the sedimentation assay (Fig. 1 C). Interestingly, kinetics of MakA binding with SLB was also pH dependent. When pH was increased from 5.4 to 6.8, the time scale of MakA binding enhanced from 10 to 200 min. Treatment with buffer (pH 7.6) could not dissociate the bound MakA from SLB, implying the interaction between MakA and membrane is not reversible by increasing pH (Fig. S2 B).

Next, we performed microscale thermophoresis (MST) analysis to quantify MakA's affinity to cholesterol in solution. NBD-labeled cholesterol (22-NBD-cholesterol) bound to MakA in solution with an estimated $K_d$ of 22 µM at pH 6.0 and 78 µM at pH 7.6 (Fig. S2 C), suggesting that MakA weakly binds to cholesterol in solution. Taken together, these results demonstrate that the kinetics of MakA binding to the membrane is pH-dependent and MakA is capable of binding cholesterol in solution.

### Structure-function analysis of MakA-cholesterol interaction on the membrane

According to previously published crystal structures (PDB accession nos. 6EZV and 6DFP; Dongre et al., 2018; Herrera et al., 2022), MakA folds into a five helical bundle structure ($\alpha 1$–$\alpha 3$, $\alpha 6$,

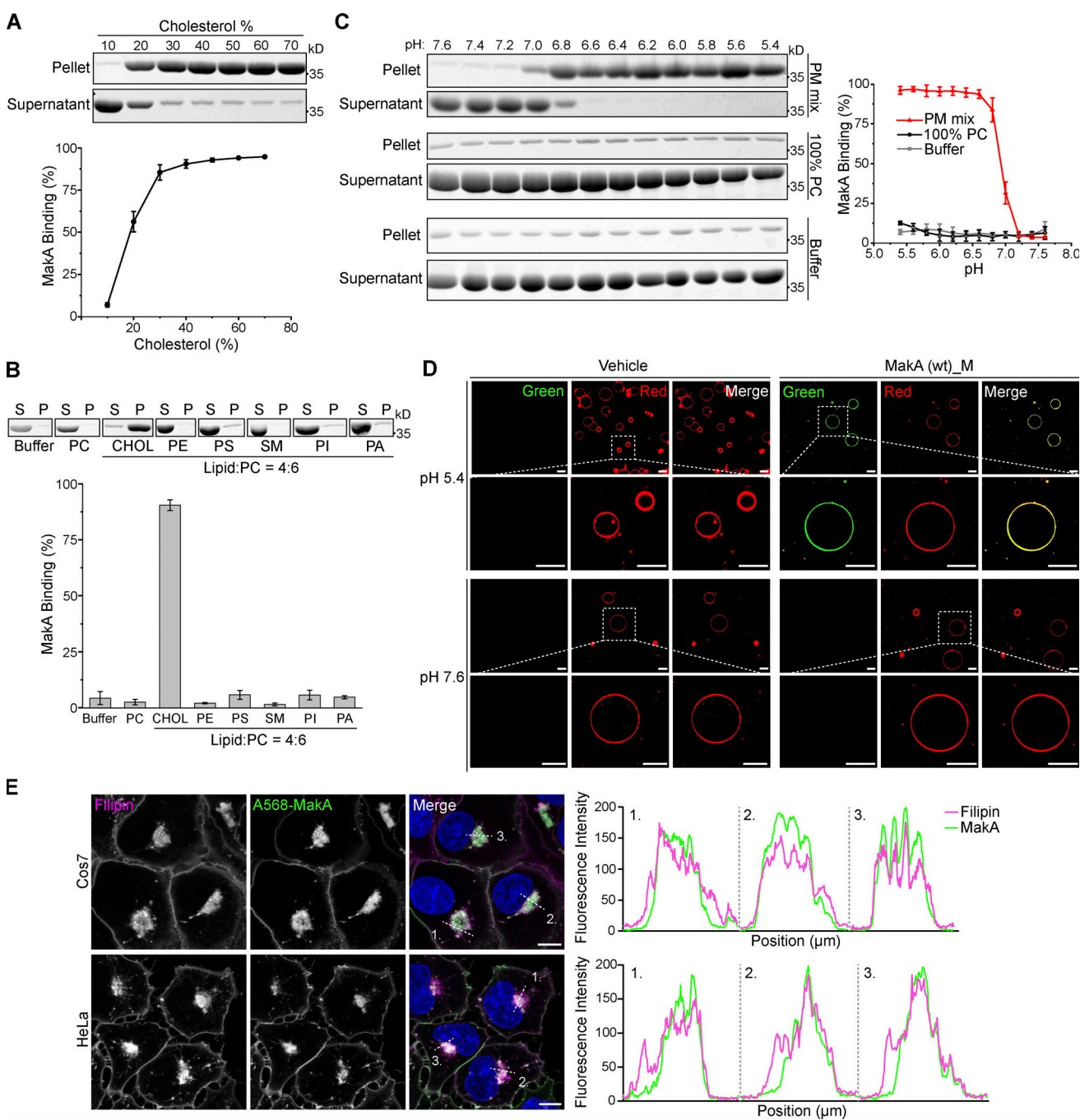

Figure 1. **MakA's specific binding to cholesterol on the membrane at pH < 7.** Wild-type monomeric MakA was used in the following assays. **(A)** SDS-PAGE and analysis of liposome sedimentation assay of at pH 6.8; the liposomes contained cholesterol and DOPC, with increasing ratio of cholesterol (10–70%). Data are shown as mean ± SD from three independent experiments. **(B)** SDS-PAGE and analysis of MakA's liposome sedimentation assay at pH 6.8; the liposomes are 100% DOPC or Lipid: DOPC (4:6). Buffer refers to 25 mM Bis-Tris (pH 6.8), 109.5 mM NaCl, 5.4 mM KCl, 0.4 mM MgSO₄, 0.45 mM CaCl₂; S: supernatant; P: pellet; PC: DOPC; CHOL: cholesterol; PE: DOPE; PS: DOPS; SM: brain sphingomyelin; PI: Soy PI; PA: DOPA. Data are shown as mean ± SD from three independent experiments. **(C)** SDS-PAGE and analysis of MakA's liposome sedimentation assay from pH 5.4–7.6; the liposomes are 100% DOPC or PM mix (50% DOPC, 20% cholesterol, 10% DOPE, 5% DOPS, 5% Soy PI, 10% Brain SM); the buffer is 25 mM Bis-Tris of the indicated pH, 109.5 mM NaCl, 5.4 mM KCl, 0.4 mM MgSO₄, 0.45 mM CaCl₂. Data are shown as mean ± SD from three independent experiments. **(D)** Representative images of GUV (POPC/POPE/cholesterol = 7/1/ 2) treated with 0.5 μM Alexa568-MakA for 150 min at pH 5.4 or 7.6; Green channel refers to Alexa568-MakA; Red channel refers to ATTO647N-DOPE. Scale bar, 50 μm. **(E)** Cos7 (top) and HeLa (bottom) cells treated with 250 nM Alexa568-MakA mutant for 16 h and subjected to filipin staining to visualize cholesterol. Nuclei were counterstained with DRAQ5. Scale bars, 10 μm. Fluorescence intensity profiles along internalized aggregates (dotted white line) are shown to the right of the corresponding image. Data are representative of >100 cells from three independent experiments. Source data are available for this figure: SourceData F1.

and α7) with an associated domain composed of two helices (α4 and α5) and three β-strands (β2, β3, and β4; Fig. 2 A). MakB (PDB accession nos. 6W1W and 6T8D) and MakE (PDB accession nos. 6W08 and 6TAO) are structurally similar to MakA with root-mean-square deviation (rmsd) of 3.0 and 2.3 Å, respectively (Fig. S3 A; Nadeem et al., 2021; Herrera et al., 2022). Therefore, we used sedimentation assays to investigate if MakB or MakE might show similar cholesterol-dependent membrane interaction. Negligible MakE or MakB was precipitated with liposomes containing 70% cholesterol under pH ranging from 5.4 to 7.6, suggesting that MakE and MakB do not bind to cholesterol on the membrane (Fig. S3 B). Consistently, no interaction was detected between 22-NBD-cholesterol and MakE or MakB by MST (Fig. S2, D and F). These results indicate that only MakA is capable of binding cholesterol, despite the fact that all three proteins are expressed in the same operon and share similar 3D structures. Thus, we hypothesized that slight differences in hydrophobic residues between MakA and MakB/MakE may determine MakA's specific interaction with cholesterol.

To explore the key hydrophobic residues of MakA responsible for MakA-cholesterol interaction, we analyzed sequences and structures of MakA, MakB, and MakE. Hydrophobicity plots generated from ProtScale indicated a hydrophobic region in head domains of MakA, MakB, and MakE (Fig. S3 C). In MakA, residues 196–246 were recognized as a hydrophobic region (score > 1.2). A hydrophobic region is located in residues 196–220 in MakB and residues 202–226 in MakE. The hydrophobic region of MakA consists of two β sheets (β2, β3), two loops (α4-β2 loop, β3-α5 loop), and the beginning of α5. In contrast, the hydrophobic region of MakB only contains half of β2 and the whole β3, while the hydrophobic region of MakE consists of β2 and most of β3 (Fig. S3, A and C). Therefore, we hypothesized that residues in the α4-β2 loop, the β3-α5 loop and the beginning of α5 may be essential for MakA-cholesterol interaction.

We mutated hydrophobic residues I236 and I237 in the β3-α5 loop and W243 at the beginning α5 (Fig. 2 A) to the hydrophilic residue Asp. Surprisingly, MakA$^{W243D}$ was 100% oligomeric (Fig. S4, C and F), as shown by gel filtration, while MakA$^{I236D\&I237D}$ was 100% monomeric (Fig. S4, B and E). However, wild-type MakA (MakA WT) displayed 20% oligomer (MakA_O) and 80% monomer (MakA_M; Fig. S4, A and D). To investigate whether oligomeric MakA and these mutants interact with cholesterol on the membrane, liposome sedimentation assays were performed for MakA$^{I236D\&I237D}$ and liposome flotation assays were performed for MakA$^{W243D}$ and MakA_O. MakA$^{I236D\&I237D}$ did not bind to the membrane (Fig. 2 B) or cholesterol in solution (Fig. S2, E and F). Consistent with the results of the sedimentation assay, MakA$^{I236D\&I237D}$ did not associate with GUVs containing cholesterol (Fig. 2 D). MakA$^{W243D}$ and MakA_O did not bind to the membrane as determined by liposome flotation assays and GUV assays (Fig. 2, C–D). In line with the biochemical results, MakA$^{I236D\&I237D}$, MakA$^{W243D}$, and MakA_O showed no sign of membrane binding in MEF cells (Fig. 2 E) and, consequently, were unable to induce cholesterol-rich aggregate formation (Fig. 2, E and F) or LC3 lipidation (Fig. 2 G).

Next, we tested function of the hydroxyl group of cholesterol in MakA-cholesterol interaction. MakA interacted with

cholesterol acetate on the membrane at lower pH, in comparison with cholesterol, indicating that the hydroxyl group of cholesterol might be somewhat involved in the MakA-cholesterol interaction (Fig. S3 D). Since I236 and I237 are essential for MakA's binding to cholesterol on the membrane, we wondered if S235 may form a hydrogen bond with the 3-hydroxyl group of cholesterol, similar to the cholesterol recognition motif (Thr-Leu pair) found in the cholesterol-dependent cytolysins (CDCs; Hotze and Tweten, 2012). However, MakA$^{S235A}$ and MakA$^{S235A\&I236A}$ displayed similar binding to cholesterol on the membrane as MakA WT, indicating that S235 is dispensable for MakA-cholesterol interaction (Fig. S3 E). Collectively, these results suggest that the Ile-Ile pair (I236 and I237) in the β3-α5 loop of MakA is essential for its interaction with cholesterol on the membrane.

## MakA assembles into pore on the membrane at low pH

MakA$^{W243D}$ and MakA$^{I236D\&I237D}$ exist as 100% oligomer and 100% monomer in solution, respectively, compared to 20% oligomer for MakA WT. This result suggests that MakA has the ability to oligomerize and that perturbation of the hydrophobic region of MakA can modulate this ability. Inspired by this result, we hypothesized that low pH and cholesterol-binding may trigger the oligomerization of MakA on the membrane. MakA has structural similarity to some PFTs, including ClyA from *E. coli* (PDB accession nos. 1QOY and 2WCD; Mueller et al., 2009; Wallace et al., 2000), HBL-B and Nhe from *Bacillus cereus* (PDB accession nos. 2NRJ and 4K1P; Ganash et al., 2013; Madegowda et al., 2008), AhlB from *A. hydrophila* (PDB accession no. 6GRK; Wilson et al., 2019), SmhA and SmhB from *S. marcescens* (PDB accession nos. 7A27 and 6ZZH; Churchill-Angus et al., 2021). To explore if MakA also forms pores, we used electron microscopy to image MakA on the membrane. Monomeric MakA was incubated with nanodiscs (PC/PE/cholesterol = 3:1:1) at pH 5.4. The MakA-nanodisc complex was purified by gel filtration. Negative-stain micrograph and 2D class averages of 4,500 cryo-electron micrographs revealed that MakA forms a propeller-like shape with a ∼3 nm pore in the center (Fig. 3 A).

Next, a liposome leakage assay was used to examine the membrane-perforating activity of MakA. MakA induced a dose-dependent leakage of unilamellar liposomes (PC/PE/cholesterol = 7:1:2) at pH 5.4 (Fig. 3 B), whereas much less leakage was observed at pH 6.8 and 7.6 (Fig. 3 C). Interestingly, MakA interacts with membrane at pH 6.8 without membrane perforation (Fig. 1 C and Fig. 3 C), suggesting that MakA exists in different structural states at different pHs. Overall, these results indicate that low pH triggers MakA to oligomerize into a pore on the membrane.

## Low pH triggers MakA-induced membrane damage and non-canonical autophagy

How a cell responds to membrane damage is determined, in large part, by the extent of the damage. Small membrane lesions (<100 nm) have been shown to preferentially trigger a membrane-repair mechanism mediated by the endosomal sorting complex required for transport (ESCRT) machinery (Jimenez et al., 2014; Radulovic et al., 2018; Scheffer et al., 2014). Given the

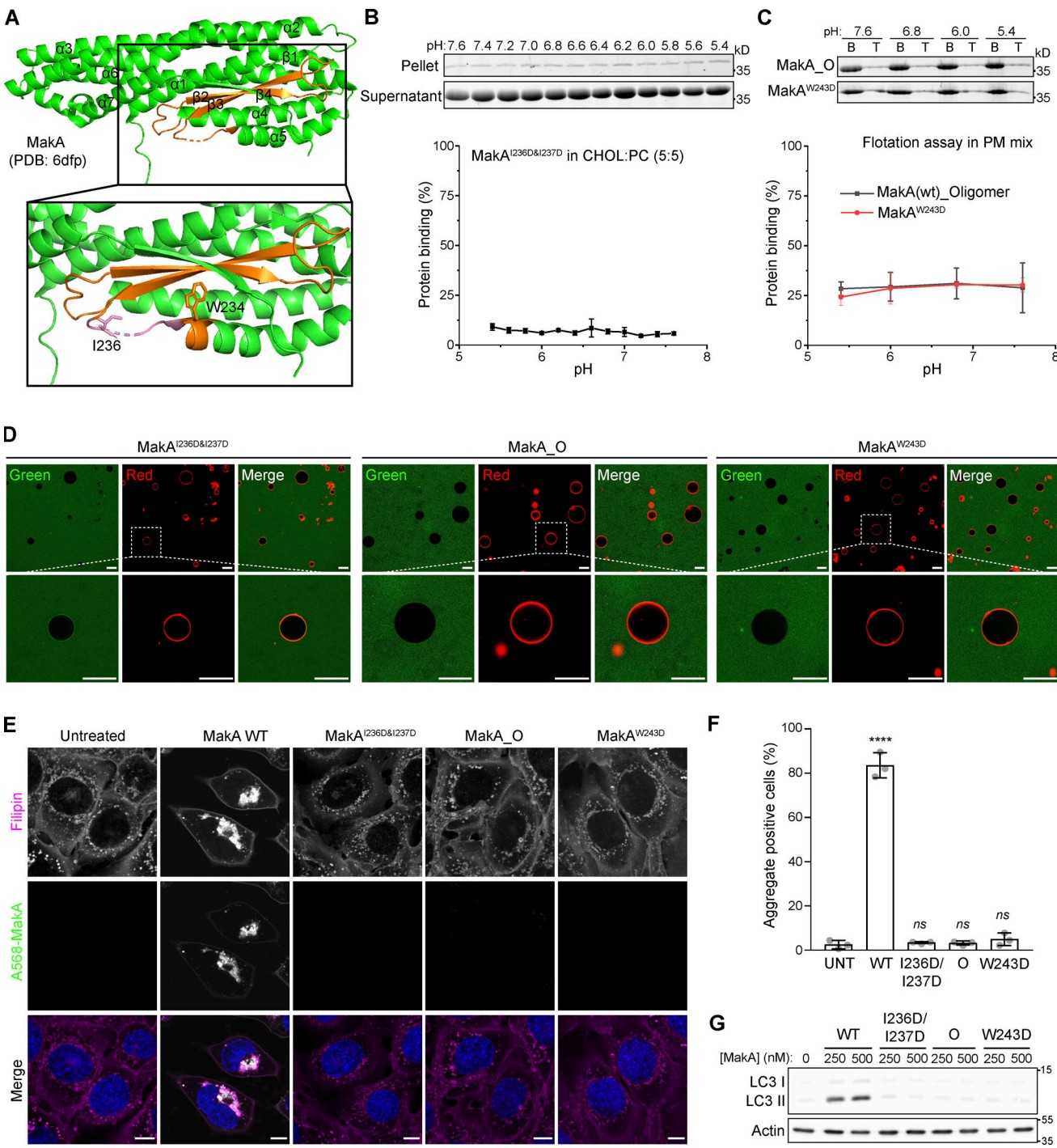

Figure 2. **β3-α5 loop is responsible for MakA's binding to the membrane and MakA's function in cells. (A)** 3D structure of MakA (PDB accession no. 6DFP), hydrophobic residues from Kyte & Doolittle plot are highlighted in orange; β3-α5 loop is colored in pink, I236 and W243 are showed with side-chain sticks. **(B)** SDS-PAGE and analysis of liposome sedimentation assay of MakA[I236D&I237D] from pH 5.4–7.6; the liposomes contain 50% cholesterol and 50% DOPC. Data are shown as mean ± SD from three independent experiments. **(C)** SDS-PAGE and analysis of liposome flotation assay of MakA_O and MakA[W243D] from pH 5.4–7.6; liposomes are PM mix (50% DOPC, 20% cholesterol, 10% DOPE, 5% DOPS, 5% Soy PI, 10% Brain SM). B: bottom fraction; T: top fraction. Data are shown as mean ± SD from three independent experiments. **(D)** Representative images of GUV (POPC/POPE/cholesterol = 7/1/2) treated with Alexa568-MakA[I236D&I237D], Alexa568-MakA_O, or Alexa568-MakA[W243D] for 30 min at pH 5.4. **(E)** MEFs treated with 250 nM of the indicated Alexa568-labeled MakA_M, MakA_O, and MakAmutants for 16 h and subjected to filipin staining to visualize cholesterol. Nuclei were counterstained with DRAQ5. Scale bars, 10 μm. **(F)** Quantification of aggregate formation from E. Bars show mean ± SD from three biologically independent experiments which are represented as data points (*n* > 115 cells per experiment). Significance was determined from biological replicates using an ordinary one-way ANOVA with Dunnett's multiple comparisons test. ****P = <0.0001. **(G)** Western blot analysis of LC3 lipidation in MEFs treated with the indicated concentration of MakA proteins for 16 h. Source data are available for this figure: SourceData F2.

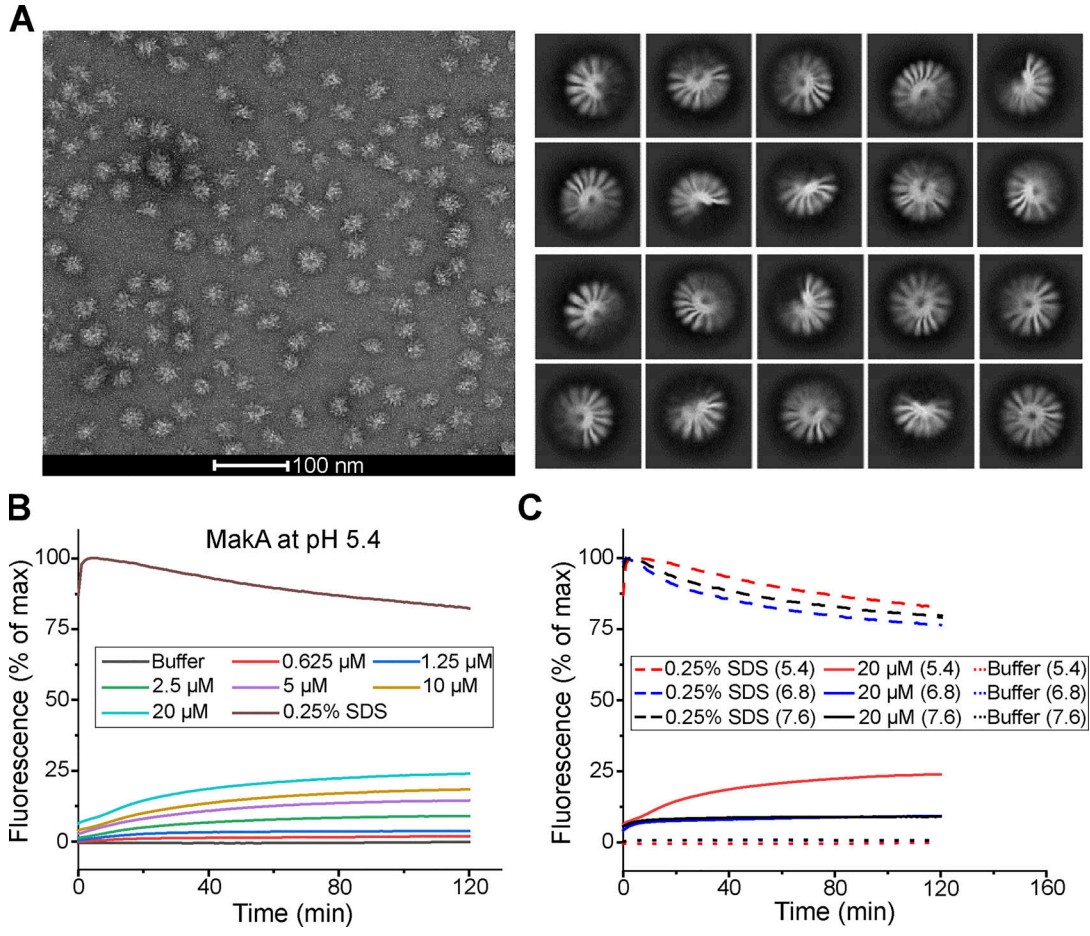

Figure 3. **MakA assembles into pores at low pH. (A)** Representative negative-stain micrograph and representative Cryo-EM 2D class averages of MakA_M that binds to nanodiscs containing DOPC/DOPE/cholesterol (3/1/1) at pH 5.4. **(B)** Dose-dependent liposome leakage induced by MakA_M at pH 5.4. Buffer refers to 25 mM Bis-Tris (pH 5.4), 109.5 mM NaCl, 5.4 mM KCl, 0.4 mM MgSO$_4$, 0.45 mM CaCl$_2$. **(C)** pH-dependent liposome leakage induced by MakA_M (20 μM). Liposomes contain DOPC/DOPE/cholesterol (7/1/2, molar ratio). Liposome leakage was monitored by measuring fluorescence of released sulforhodamine-B, and 0.25% SDS treatment was used as a positive control of 100% dye leakage. Buffer refers to 25 mM Bis-Tris of the indicated pH, 109.5 mM NaCl, 5.4 mM KCl, 0.4 mM MgSO$_4$, 0.45 mM CaCl$_2$.

small size of MakA pores observed in vitro (Fig. 3 A), we hypothesized that similar lesions in cells would be detected and repaired by the ESCRT machinery. To test this hypothesis, HeLa cells (Fig. 4) and mouse embryonic fibroblasts (MEFs; Fig. S5 A) were treated with Alexa568-MakA for 16 h and immunostaining performed against ESCRT machinery components. ESCRT-III complex members IST1 and CHMP2A, as well as the ESCRT-III-binding protein, ALIX, were all recruited to MakA-aggregates suggesting activation of ESCRT-mediated membrane repair.

If membrane damage is too extensive to be repaired through the ESCRT machinery, then the cell engages autophagy-dependent removal of the damaged membranes. Extensively damaged cellular membranes are sensed by the Galectin family of carbohydrate-binding proteins. Recruitment of Galectins to damaged membranes acts as a signaling platform to induce autophagic degradation (Chauhan et al., 2016). In agreement with previous results (Corkery et al., 2021), we observed no Galectin 3 or Galectin 8 recruitment to MakA aggregates (Fig. S5 B), indicating MakA-induced membrane damage is not extensive

enough to engage the autophagy-dependent membrane removal pathway.

During endocytosis, endosomal pH decreases from ~7.2 (cytoplasmic pH) to ~6.5 (early endosome pH), then down to ~5.5 (late endosome pH) by the proton pumping ability of vacuolar-type H$^+$-ATPase (V-ATPase; Hu et al., 2015). This pH range corresponds to the slightly acidic conditions in which we observed MakA interaction with cholesterol on the membrane as well as the low pH that was required for MakA's pore-forming activity. To explore the relationship between pH-dependent MakA pore-forming activity, membrane-damage, and MakA-induced non-canonical autophagy, a panel of autophagy-deficient cell lines were treated with the V-ATPase inhibitor Bafilomycin A1 (BafA1) to prevent endolysosomal acidification. In wild-type cells, BafA1 treatment leads to an accumulation of LC3 II due to inhibition of basal autophagic flux (Fig. 5, A and B). MakA treatment, as described previously, leads to an accumulation of LC3 II due to unconventional LC3 lipidation on endolysosomal membranes (Corkery et al., 2021). Co-treatment with both BafA1 and MakA does not have an additive effect on LC3 II

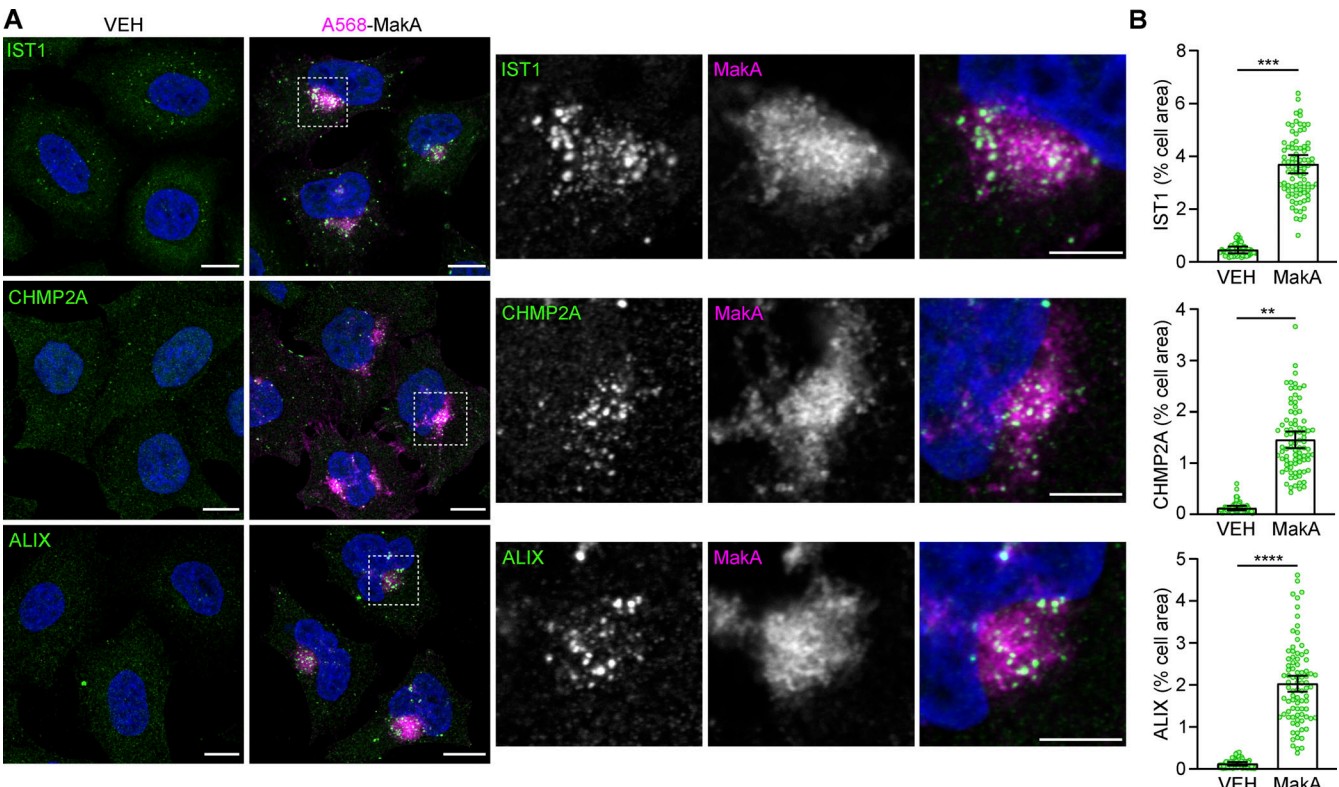

**Figure 4. The ESCRT machinery is recruited to MakA-induced aggregates. (A)** HeLa cells treated with 125 nM Alexa568-MakA (monomeric) for 16 h and co-stained for endogenous IST1, CHMP2A or ALIX, as indicated. Nuclei were counterstained with DAPI. Scale bars = 10 μm for whole images (left) and 5 μm for insets (right). **(B)** Quantification of the percentage of cell area occupied by IST1, CHMP2A, or ALIX from A. Bars show mean ± SD from three biologically independent experiments. Data points represent individual cells pooled from the three independent experiments ($n > 25$ cells per experiment). Significance was determined from biological replicates using a two-tailed, unpaired $t$ test. **P = 0.0002, ***P = 0.0001, ****P < 0.0001.

accumulation, as would be anticipated with the induction of two separate pools of lipidated LC3. One possible explanation is that MakA-induced LC3 lipidation is dependent on endosomal acidification. To confirm, we performed the same experiment in cells deficient for basal autophagy (FIP200KO and ATG9KO). In these cells, BafA1 alone does not induce LC3 II accumulation due to an absence of autophagic flux. MakA, as an inducer of non-canonical autophagy, is still able to induce LC3 lipidation in the absence of basal autophagy. Importantly, co-treatment with MakA and BafA1 in cells deficient for basal autophagy completely blocks MakA-induced LC3 lipidation (Fig. 5, A and B) confirming inhibition of MakA-induced LC3 lipidation by BafA1.

While our data would suggest that the observed inhibition of LC3-lipidation by BafA1 could be due to an inhibition of pore formation, V-ATPase has also been shown to play a direct role in regulating non-canonical LC3 lipidation through recruitment of the ATG5-ATG12-ATG16L1 E3-like conjugation complex (Ulferts et al., 2021). To determine if BafA1 treatment was simply inhibiting LC3 lipidation, we performed immunostaining for the ESCRT component ALIX in MakA- and BafA1-treated cells (Fig. 5 C). In WT cells, co-treatment with MakA and BafA1 inhibited assembly of MakA-induced endolysosomal membrane aggregates (Fig. 5, C and D) and reduced ALIX accumulation confirming inhibition of membrane damage. Aggregate assembly was also not blocked in cells deficient for LC3

lipidation (ATG7KO; Fig. 5 E), further confirming reduced aggregate assembly is not a consequence of BafA1-mediated LC3 lipidation inhibition. Furthermore, the expression of the *Salmonella* T3SS effector SopF, which has been shown to inhibit ATG16L1 recruitment to V-ATPase without inhibiting endosomal acidification (Xu et al., 2022; Xu et al., 2019), prevented LC3 lipidation, but did not prevent aggregate assembly or recruitment of the ESCRT machinery (Fig. S5 C).

In WT, basal autophagy-deficient (FIP200KO) or LC3 lipidation-deficient (ATG7KO) cells, MakA treatment alone produced aggregates that recruit the ESCRT machinery, independent of LC3. Co-treatment with BafA1 inhibited aggregate assembly and reduced ESCRT machinery recruitment (Fig. 5, C–E). Collectively, these data support our hypothesis that endosomal acidification induces MakA pore formation leading to V-ATPase-dependent non-canonical LC3 lipidation on damaged endolysosomal membranes.

## Discussion

MakA was shown previously to induce aggregate formation and LC3 lipidation in host cells within a narrow pH window (6.8–7.4; Corkery et al., 2021). Our work expands upon this finding by characterizing pH-dependent interactions between MakA and cholesterol-rich membranes. At pH 6.8 MakA binds, but does

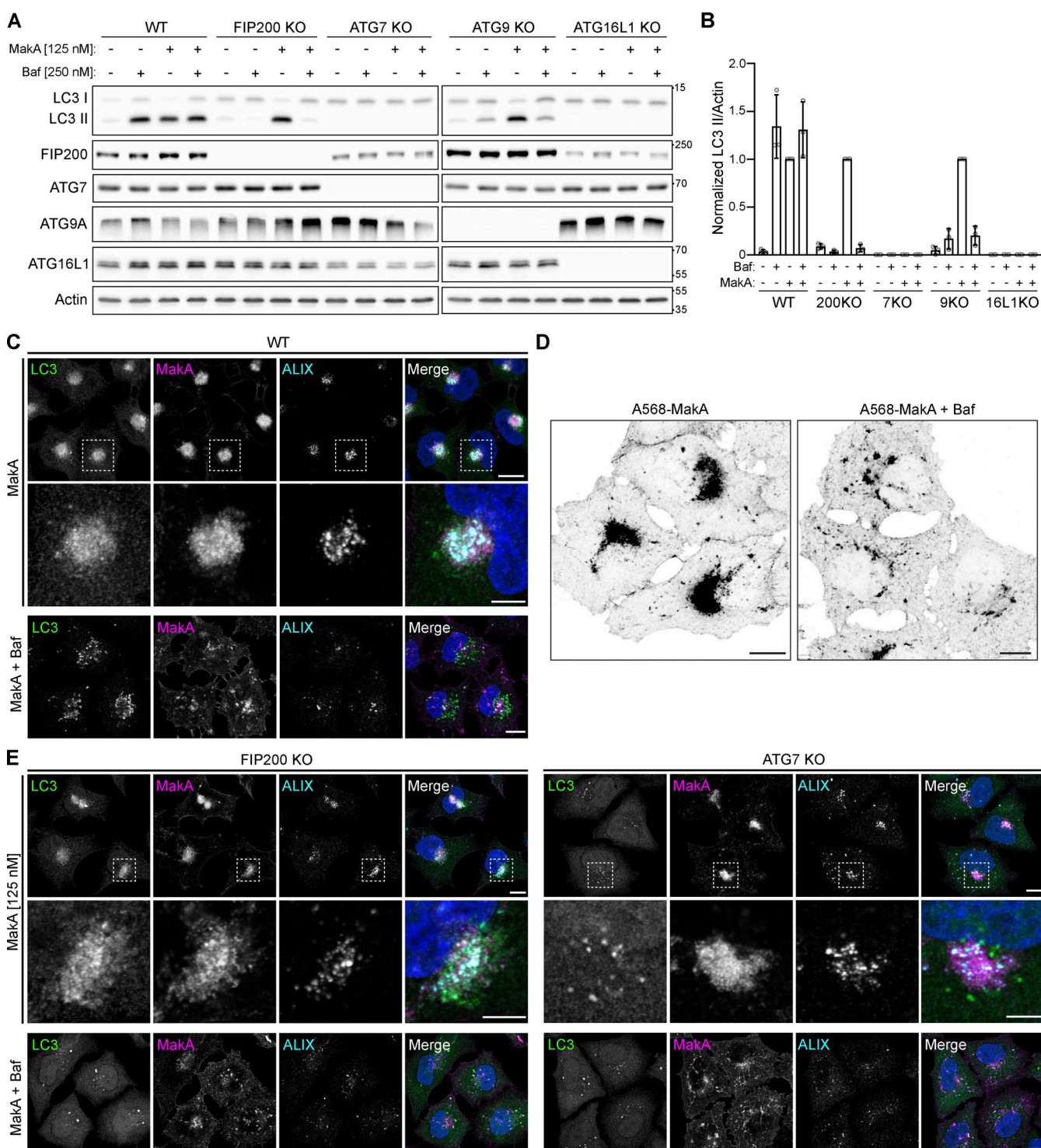

**Figure 5. Inhibition of endosomal acidification prevents MakA-induced aggregate formation and the induction of non-canonical autophagy.** Alexa568-MakA refers to Alexa568 labeled monomeric MakA. **(A)** Western blot analysis of LC3 lipidation status in WT and a panel of autophagy-deficient HeLa cells treated with 125 nM MakA_M and/or 250 nM Baf for 16 h, as indicated. **(B)** Quantification of LC3 II levels from immunoblots in A, normalized to MakA_M-treated cells. Bars show mean ± SD from three biologically independent experiments which are represented as data points. **(C)** Representative confocal images of WT HeLa cells treated with 125 nM Alexa568-MakA with or without 250 nM Baf for 16 h and immunostained for LC3 and ALIX. Nuclei were counterstained with DAPI. Scale bars = 10 μm for whole images and 5 μm for insets. **(D)** Inverted fluorescent images of Alexa568-MakA accumulation in WT HeLa cells treated with 125 nM Alexa568-MakA with and without Baf for 16 h. Scale bars, 10 μm. **(E)** Representative confocal images of FIP200 (left) and ATG7 (right) deficient HeLa cells treated with 125 nM Alexa568-MakA with or without 250 nM Baf for 16 h and immunostained for LC3 and ALIX. Nuclei were counterstained with DAPI. Scale bars = 10 μm for whole images and 5 μm for insets. Source data are available for this figure: SourceData F5.

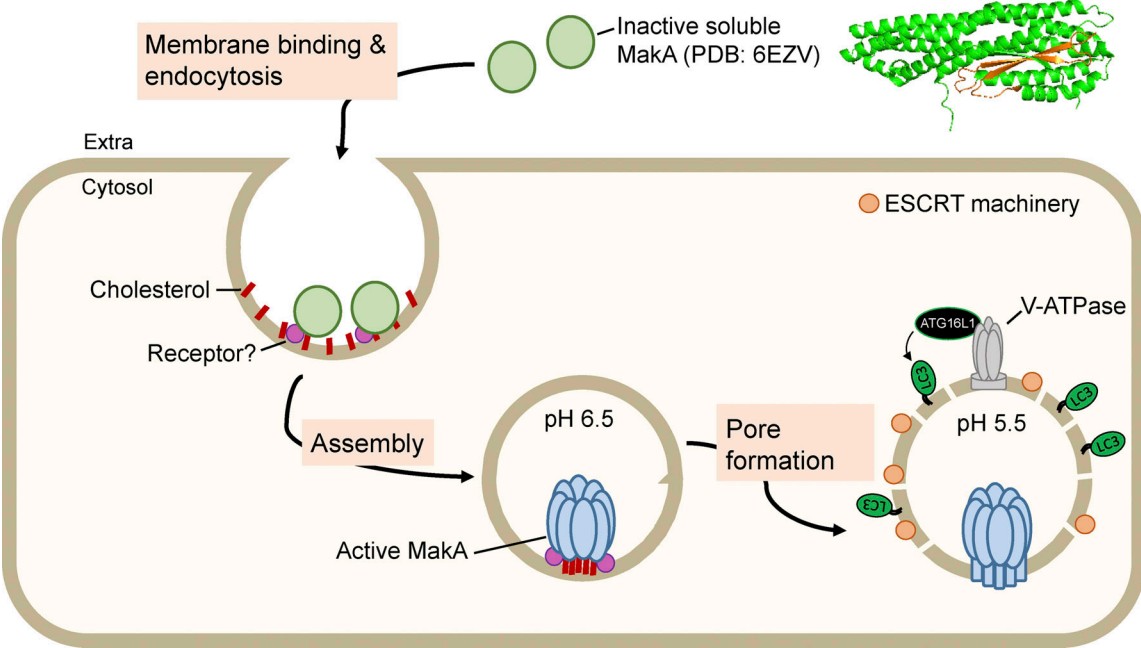

Figure 6. **MakA assembles into a pH-dependent pore during endocytosis.** MakA binds the plasma membrane and is internalized into the cell via endocytosis. Endosomal acidification promotes pre-pore assembly in early endosomes (pH 6.5). Further acidification promotes pore formation in late endosomes/lysosomes (pH 5.5) leading to membrane damage, ESCRT machinery recruitment and activation of the V-ATPase-ATG16L1 axis and unconventional LC3 lipidation on the damaged membrane.

not perforate, cholesterol-rich membranes. At pH 5.4, MakA inserts into the membrane to form a pore (Fig. 3). This suggests that MakA can alter its state in response to environmental pH changes. This oligomerization process on membranes may be triggered by the weak MakA-cholesterol interaction at neutral pH (7.0–7.2; Fig. 1 C) and/or unidentified cell-surface receptors with a reduction in pH facilitating multivalent binding and increased affinity for the membrane. Interestingly, the pH range in which we observed a change in MakA states in vitro (7.0–5.4) aligns with the pH range observed during endosomal acidification. Thus, we proposed that MakA assembles into a pre-pore state to be internalized into the cell, followed by assembly into a pore state at ca. pH 5.5 in late endosomes/endolysosomes (Fig. 6).

This model is consistent with our findings in cells. We found that MakA-induced endolysosomal membrane aggregates are targeted by the ESCRT machinery, suggesting endolysomal membrane damage caused by MakA (Fig. 4). Moreover, BafA1 treatment reduced aggregate assembly and ESCRT recruitment independent of LC3 lipidation, suggesting that MakA-induced endolysosomal damage is correlated with pH-dependent MakA pore formation (Fig. 5). MakA-induced endolysosomal damage signals through the V-ATPase-ATG16L1 axis to induce non-canonical LC3 lipidation similar to what has been reported for the PFT listeriolysin O (LLO) from *Listeria monocytogenes*, which induces non-canonical LC3 lipidation by LLO-inflicted damage to the phagosomal membrane (Gluschko et al., 2022). While CDCs usually form large ring-like structures with a central pore diameter of ∼20 nm (Iacovache et al., 2008), MakA assembles into a much smaller pore (∼3 nm). A pore of this size would allow for

the passage of ions or small molecules but not most folded proteins, which is consistent with our observation that MakA induces small ruptures in the membrane (recruitment of the ESCRT machinery, Fig. 4), but not large ruptures (negative for Galectin3/8, Fig. S5 B).

Cholesterol interaction is the hallmark of the CDC family with a conserved cholesterol-binding motif (Thr-Leu pair) identified (Farrand et al., 2010). A plausible binding mechanism of CDCs has been proposed whereby the leucine inserts into the lipid bilayer and interacts with hydrophobic rings of cholesterol. The threonine then forms a hydrogen bond with the 3-hydroxyl group of cholesterol (Hotze and Tweten, 2012). Unlike the traditional CDC cholesterol-binding motif, we identified an Ile-Ile pair (I236 and I237) in MakA essential for its interaction with membrane cholesterol. The adjacent S235 is dispensable for this interaction, yet the 3-hydroxyl of cholesterol appears to be somewhat important. This suggests that MakA features a novel cholesterol-binding mechanism different from other CDCs and in keeping with the notion that MakA displays low structure similarity with CDCs. While cholesterol interaction is believed to serve as a common receptor amongst CDCs, it has been reported that some CDCs utilize additional receptors to carry out their function in host cells. For example, Intermedilysin (ILY) from *Streptococcus intermedius* uses human CD59 (huCD59) as a cellular receptor (Giddings et al., 2004), although cholesterol is still required for membrane insertion of the ILY pore (Boyd et al., 2016). A recent study has further shown that all major CDCs bind glycans and cholesterol independently and use glycans as cellular receptors to regulate cell tropism (Shewell et al., 2020; Shewell et al., 2014). Therefore, while it is clear that cholesterol

is important for the interaction between MakA and cellular membranes, we cannot exclude that MakA may have additional binding partners at the cell surface.

In earlier studies, several PFTs have been identified showing acidic pH-induced membrane insertion, such as anthrax toxin from *Bacillus anthracis* (Jiang et al., 2015), colicin A from *Escherichia coli* (Davidson et al., 1985), diphtheria toxin (DT) from *Corynebacterium diphtheria* (Rodnin et al., 2016), LLO (Schuerch et al., 2005), vacuolating cytotoxin A (VacA) from *Helicobacter pylori* (de Bernard et al., 1995), perfringolysin (PFO) from *Clostridium perfringens* (Nelson et al., 2008), lysenin from earthworm *Eisenia foetida* (Munguira et al., 2017), Tripartite toxin complexes (Tc) from a variety of insect and human pathogens (Meusch et al., 2014), and a recently reported mammalian pore-forming protein perforin-2 (PFN2; Ni et al., 2020). Most of these PFTs undergo a pre-pore to pore transition under low pH with plausible pH-sensitive domains reported. The protonation of three glutamic acids in a short α-helix of lysenin were identified to be involved in low pH-induced membrane insertion (Munguira et al., 2017). For anthrax toxin, it was demonstrated that one $2\beta_{10}$–$2\beta_{11}$ loop (N422, D425) served as a pH sensor to trigger conformation changes for pre-pore to pore conversion (Jiang et al., 2015). The putative pH-sensing mechanism in MakA is still unclear, but likely relates to protonation of specific amino acids under low pH. In addition to the closed pore formation of PFTs, oligomerization occasionally ends up with an incomplete pore-like arc-shape. Using real-time atomic force microscopy (AFM), cryo-EM, and atomic structure fitting, it was demonstrated that suilysin assembly resulted in not only ring-shaped pores but also kinetically trapped arc-shaped oligomers, both capable of perforating the membrane (Leung et al., 2014). Arc-shaped structures of LLO were also confirmed using high-speed AFM, which revealed arc pores inserted into membrane causing damage (Ruan et al., 2016). Membrane attack complex (MAC), an immune system PFT, was also shown to form pores in a closed and open conformation (Menny et al., 2018).

The crystal structure of MakA monomer is similar to toxins in the ClyA α-PFT family (Churchill-Angus et al., 2021; Dongre et al., 2018; Ganash et al., 2013; Herrera et al., 2022; Madegowda et al., 2008; Mueller et al., 2009; Wilson et al., 2019). Recently, a tubular structure of MakA oligomer at pH 6.5 showed that MakA undergoes conformational changes from a soluble to a membrane-bound state (Nadeem et al., 2022). The propeller-shape of the 2D averages of MakA (Fig. 3 A) is similar in dimension and organization to the top view of the tubular structure. The Ile-Ile pair (I236 and I237) essential for cholesterol interaction is located within one of the lipid interaction regions (222–242) in the tubular structure (Nadeem et al., 2022; Fig. 7 B). It is not yet clear if the MakA tubes form only under specific conditions in vitro or if they have certain physiological functions. However, the similar propeller shape of 2D top views of the pore and the tube suggests that the tubular structures may somewhat reflect the organization of the pore and the conformational changes during pore formation.

Earlier studies have shown that other PFTs from the ClyA family (ClyA, HBL-B and Nhe) also require cholesterol for their insertion into membranes (Dal Peraro and van der Goot, 2016). Cholesterol stimulates ClyA pore formation by a dual-mode of selectively stabilizing a protomer-like conformation and bridging interactions in the protomer–protomer interfaces. Tyr27 in the N-terminal helix of ClyA is shown to be a key determinant for ClyA-cholesterol interaction, whereas the cholesterol-interacting Ile-Ile pair of MakA is located within the β-tongue in the monomeric state (Dongre et al., 2018; Nadeem et al., 2022; Sathyanarayana et al., 2018). Upon membrane-binding, the N-terminal helix of ClyA folds out to insert into the membrane, where cholesterol stabilizes the conformation and the protomer–protomer interactions (Fig. 7 A). In contrast, according to the tubular structure model of MakA, the β-tongue together with α4 and α5 detaches from the core of the protein and forms two extended helices (Fig. 7 B). Whether or not cholesterol plays a similar role in MakA pore formation as it does in ClyA pore formation has yet to be determined, but the pH-dependent pore-forming activity of MakA appears to be unique amongst the ClyA α-PFTs.

## Materials and methods

### Plasmids, protein expression, and purification

The plasmid of wild-type MakA was a gift from Karina Persson (Umeå university, Umeå, Sweden). mCherry-SopF was a gift from Feng Shao (National Institute of Biological Sciences, Beijing, China; Xu et al., 2019). MakA mutants were generated with Agilent QuikChange II Site-directed Mutagenesis Kit and verified by DNA sequencing. Primers used for MakA mutants are listed in Table 1.

Wild-type MakA and MakA mutants were expressed in *E. coli* BL21 (DE3) in 4 liters of LB broth. Induction was initiated at $OD_{600}$ (0.5–0.7) with 0.5 mM Isopropyl β-d-1-thiogalactopyranoside (IPTG) followed by overnight growth at 20°C (wild-type MakA) or 25°C (MakA mutants). Cells were harvested by centrifugation at 4°C, 6,000 rpm for 15 min. Discard supernatant and pellets were stored at –80°C until further use.

The cell pellets from 4 liters of cell culture were combined and resuspended with breaking buffer (50 mM Tris, 300 mM NaCl, 10 mM imidazole, 2 mM β-mercaptoethanol, pH 8.0) freshly supplemented with 100 mM phenylmethylsulfonyl fluoride (PMSF) and 0.1% antifoam. Suspended cells were lysed by Cell Disruption (Constant Systems). The lysate was supplemented with 1% Triton X-100 and centrifuged at 4°C, 6,000 rpm for 20 min. Supernatant was filtered by 0.2 μm membrane and then loaded onto a 5-ml HisTrap HP (Cytiva) column. The column was equilibrated with breaking buffer, and protein was eluted with a step imidazole gradient (100 mM imidazole for 25 ml, 200 mM imidazole for 25 ml, 300 mM imidazole for 50 ml, and 300–500 mM imidazole for 50 ml in gradient mode). The eluted fractions were pooled together and incubated with 1% (w/w) TEV protease overnight at room temperature (r.t.) by dialysis against buffer (50 mM Tris, 2 mM β-mercaptoethanol, pH 8.0). The cleaved protein was purified in a 5-ml HisTrap HP (Cytiva) column again to remove his-tag. The flow through was concentrated with 10kD Amicon Ultra centrifuge filter to ~10 ml. The protein was further applied to a size-exclusion chromatography

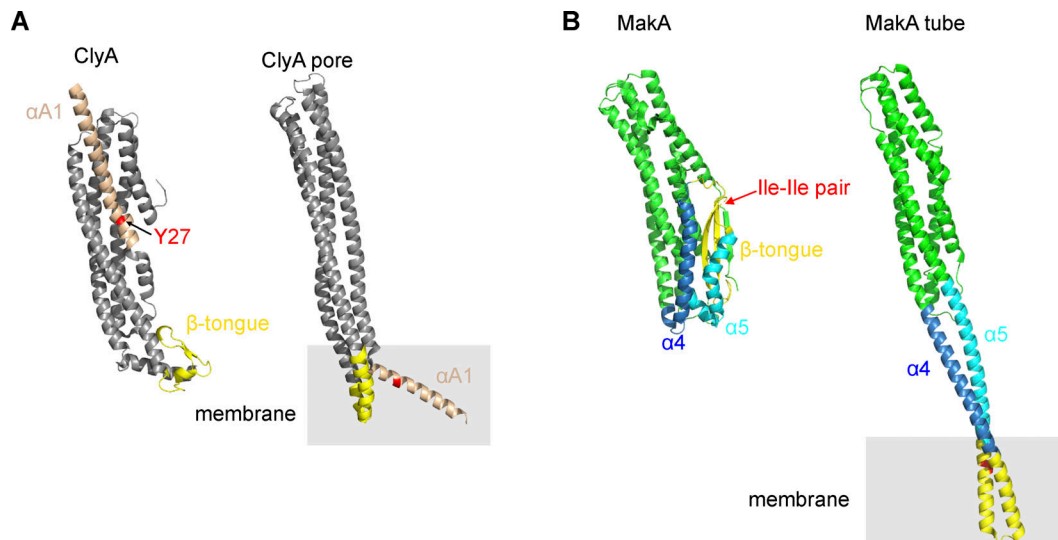

Figure 7. **Comparison between ClyA and MakA structures. (A)** Crystal structures of ClyA monomer (PDB accession no. 1QOY; Wallace et al., 2000; left) and single protomer from the ClyA pore (PDB accession no. 2WCD; Mueller et al., 2009; right). The membrane interacting domains, β-tongue (yellow), the N-terminal helix (αA1, orange) and the key cholesterol-binding residue Y27 (red), are highlighted. **(B)** Structures of MakA monomer (PDB accession no. 6EZV; Dongre et al., 2018) and a subunit of the MakA tube (PDB accession no. 7P3R; Nadeem et al., 2022). The regions that undergo conformational changes are highlighted: β-tongue (yellow), α4 (blue) and α5 (light blue). The cholesterol-interacting Ile-Ile pair is shown in red.

column (Superdex 200 pg, HiLoad 26/60) equilibrated with buffer (20 mM Tris, 200 mM NaCl, pH 7.6). Peak corresponding to monomer species or oligomer species was concentrated, respectively, and exchanged to buffer (20 mM Tris, 50 mM NaCl, pH 7.6). Proteins were aliquoted into PCR tube, flash-frozen in liquid nitrogen, and stored at –80°C until further use.

## Lipid preparation

DOPC ([1,2-dioleoyl-sn-glycero-3-phosphocholine], 850375P), DOPE ([1,2-dioleoyl-sn-glycero-3-phosphoethanolamine], 850725P), DOPS ([1,2-dioleoyl-sn-glycero-3-phospho-L-serine], 840035P), Soy PI ([L-α-phosphatidylinositol (Soy)], 840044P), Brain SM ([Sphingomyelin (Brain, Porcine)], 860062P), POPC ([1-palmitoyl-2-oleoyl-glycero-3-phosphocholine], 850457P), POPE ([1-palmitoyl-2-oleoyl-sn-glycero-3-phosphoethanolamine], 850757P), and 18:1 NBD-PE ([1,2-dioleoyl-sn-glycero-3-phosphoethanolamine-N-(7-nitro-2-1,3-benzoxadiazol-4-yl)], 810145C) were purchased from Avanti Polar Lipids Inc. Cholesterol (C8667) and cholesteryl acetate (151114) were purchased from Merck-Sigma. ATTO647N labeled DOPE (ATTO647N-DOPE, AD 647N-161) was purchased from ATTO-TEC GmbH. NBD-PE was prepared as 1 mM in chloroform, ATTO647N-DOPE was prepared as 0.1 mg/ml in chloroform, and other lipids were

prepared as 5 or 10 mM in chloroform. Store all lipid chloroform stock at –20°C.

## Liposome sedimentation assay

Multi-lamellar liposomes with indicated compositions (molar ratio) were prepared for liposome sedimentation assay. Briefly, chloroform stock (5 mM) of different lipids were mixed in a clean glass vial, dried with gentle nitrogen gas flow for 10 min, and then dried overnight in a freeze-dryer. The lipid film was rehydrated to a final concentration of total lipid (5 mM) by adding liposome buffer (25 mM Bis-tris, 109.5 mM NaCl, 5.4 mM KCl, 0.4 mM MgSO$_4$, 0.45 mM CaCl$_2$) with according pH. Rehydrate the lipid film at r.t. for 1 h with vigorous vortex in between. The resulting mixture is multi-lamellar liposomes which were stored at 4°C and used within 48 h. The multi-lamellar liposome (40 µl) was incubated with 0.5 µl protein (400 µM) at r.t. for 1 h or 10 min. The protein incubated with only liposome buffer was served as negative control. The mixtures of protein and liposome were centrifuged at 4°C, 12,700 rpm for 1 h. The supernatant and the pellet were separated carefully and analyzed by SDS-PAGE (sodium dodecyl sulfate–polyacrylamide gel electrophoresis). To load total amount of supernatant and pellet in SDS-PAGE gel, supernatant was heated at 95°C with 1X

Table 1.  **Primers used for MakA mutants**

| Name | Primer |
| --- | --- |
| MakA$^{I236D \& I237D}$ | Fwd: 5'-GTCGGCGCGGCATCGGATGATGCTGGTGGCGTTACATG-3'<br>Rev: 5'-CATGTAACGCCACCAGCATCATCCGATGCCGCGCCGAC-3' |
| MakA$^{W243D}$ | Fwd: 5'-CGATTATTGCTGGTGGCGTTACAGACGGCGTATTACAAAATCAAATTG-3'<br>Rev: 5'-CAATTTGATTTTGTAATACGCCGTCTGTAACGCCACCAGCAATAATCG-3' |

Laemmli buffer until about 10 µl. The pellet was resuspended with 10 µl 1X Laemmli buffer. After electrophoresis, the gel was stained with coomassie blue and imaged by Bio-rad Chemidoc. Images were analyzed by ImageJ software. Each liposome sedimentation assay was performed at least three times.

## Liposome flotation assay

Uni-lamellar liposomes (DOPC/cholesterol/DOPE/DOPS/Soy PI/Brain SM/NBD-PE = 50/20/10/5/5/10/1 in molar ratio) were prepared for liposome flotation assay. Uni-lamellar liposomes (final concentration of total amount of lipid as 5 mM) were prepared from multi-lamellar liposomes (same process in sedimentation assay) followed by 5 freeze-thaw cycles and extrusion through 400 nm polycarbonate membrane (Whatman) 11 times using an extruder (Avanti Polar Lipids Inc.). To make unbound protein visible, 2 µl Alexa568 labeled protein (30 µM) was supplemented into mixture with 400 nm liposomes and 2 µl unlabeled protein (400 µM) in a 0.5 ml ultraclear ultracentrifuge tube (Beckman Coulter). The resulting mixture was incubated at r.t. for 1 h and then made to 200 µl with 70% (w/v) sucrose in liposome buffer with according pH. Above this liposome mixture, 100 µl of 50% sucrose (w/v in liposome buffer with according pH) was overlaid, followed by 100 µl of 35% sucrose (w/v in liposome buffer with according pH) and 100 µl liposome buffer with according pH. Samples were centrifuged at 90,000 rpm for 3 h at 4°C in a Beckman Coulter with TLA-100.2 rotor. After centrifugation, 200 µl of fraction was taken from the top of the tube, 100 µl in the middle was discarded, and another 200 µl fraction was taken from the bottom. 40 µl of bottom and top fraction were analyzed by SDS-PAGE. Each liposome flotation assay was performed at least three times.

## Liposome leakage assay

Uni-lamellar liposomes (DOPC/cholesterol/DOPE = 7/2/1 in molar ratio) containing Sulforhodamine B were prepared for liposome leakage assay. Briefly, the dried lipid mixture (same process in sedimentation assay) was rehydrated with 0.45-µm filtered liposome buffer containing 50 mM sulforhodamine B acid to a final concentration as 5 mM total amount of lipid. The resulting mixture was incubated at 45°C for 1 h with vigorous vortex in between, followed by five freeze–thaw cycles and extrusion through 400 nm polycarbonate membrane (Whatman) 11 times using an extruder (Avanti Polar Lipids Inc.). The liposomes were used immediately or stored at 4°C and used in 24 h. The buffer exchange through NAP-10 column (Cytiva) was performed to remove residual Sulforhodamine B outside liposome. Check the fluorescence intensity of elute by Chemidoc (Bio-rad), perform buffer exchange again if it is necessary. Liposomes after buffer exchange can only be used in 10 h. To perform liposome leakage assay, dilute the dye-encapsulated liposomes 50 times with liposome buffer at according pH. The fluorescence of sulforhodamine B acid was measured in a plate reader with excitation filter of 540/25 nm and emission filter of 620/40 nm. 195 µl diluted liposomes were added into microplate wells and the fluorescence emission were recorded as $F_0$, 5 µl of protein was added into wells, and the fluorescence emission was

continuously measured as $F_{tn}$ every 1 min for 120 min. Addition of 5 µl of 10% SDS to diluted liposomes was treated as a positive control, while addition of buffer at according pH was treated as a negative control. The highest fluorescence emission from the positive control was measured as $F_{100}$. The percentage of liposome leakage is defined as: Leakage (%) = $(F_{tn}-F_0)/(F_{100}-F_0) \times 100$. Each liposome leakage assay was performed at least three times.

## Preparation and imaging of GUVs

GUVs were prepared by electron formation method. Lipid mixtures (0.42 µmol in total, POPC/POPE/CHOL = 7/1/2 in molar ratio) were dissolved into 90 µl of chloroform. ATTO647N-DOPE (1,2-Dioleoyl-sn-glycero-3-phosphoethanolamine labeled with Atto 647N) was added in the mixture to a final concentration of 5 µg/ml to fluorescently label the membranes. The lipid mixtures were spread on the conductive side of a pair of indium tin oxide (ITO) slides by three times (30 µl each time) and dried under vacuum overnight. Electron formation was carried out in 5 ml of 600 mM sucrose solution between the two conductive ITO slides under amplitude of 2 V, frequency of 100 Hz for 1 h at 45°C. Temperature was slowly decreased to 25°C in 2 h and then GUVs were ready to be used in 10 h.

SUVs (50 nm uni-lamellar vesicles, 100% POPC) were prepared for pre-coating slides (ibidi 8-well chamber slides) for imaging GUVs. Briefly, POPC lipid film was rehydrated with GUV buffer-7.6 (40 mM Tris, 280 mM NaCl, pH 7.6) or GUV buffer-5.4 (20 mM sodium acetate, 280 mM NaCl, pH 5.4) for 1 h at r.t. and then extruded 11 times through 50 nm polycarbonate membrane. Each well was coated with 150 µl of 1 mM SUVs plus 1.5 µl of 200 mM MgCl$_2$ for at least 30 min. For imaging GUVs, 75 µl of GUVs were mixed with 75 µl of GUV buffer and Alexa568 labeled protein (final concentration of 0.5 µM) and incubated at r.t. for 30 min. The according GUV buffer was used as a negative control instead of proteins. Images were recorded on a Leica SP8 FALCON confocal microscope with a 20× air objective. ATTO647N-DOPE and Alexa568 labeled proteins were excited with 577 and 646 nm lasers.

## Antibodies and reagents

Antibodies used in this study were from the following sources: LC3B (#2775, rabbit, WB: 1:1,000), FIP200 (#12436, rabbit, WB: 1:1,000), ATG7 (#8558, rabbit, WB: 1:1,000), ATG9A (#13509, rabbit, WB: 1:1,000), ATG16L1 (#8089, rabbit, WB: 1:1,000), and Gal3 (#87985, rabbit, IF: 1:400) were purchased from Cell Signaling Technology. Anti-beta-actin antibody (A2228, mouse, WB: 1:10,000) was purchased from Sigma-Aldrich. LC3 (PM036, rabbit, IF: 1:500) antibody was purchased from MBL International. ALIX antibody (634502, mouse, IF: 1:200) was purchased from BioLegend. IST1 (51002-1-AP, rabbit, IF: 1:100) and CHMP2A (10477-1-AP, rabbit, IF: 1:199) antibodies were purchased from proteintech. Gal8 antibody (NBP2-75501, rabbit, IF: 1:400) was purchased from Novus Biologicals. Goat anti-rabbit-HRP (Cat# 31460, WB: 1:10,000) and goat anti-mouse-HRP (Cat# 31430, WB: 1:10,000) antibodies were purchased from Thermo Fisher Scientific. Alexa Fluor 488/568/647-conjugated secondary antibodies for immunofluorescence were purchased from Thermo Fisher Scientific.

Reagents used in this study were from the following sources: Filipin (F9765) from Sigma-Aldrich; Bafilomycin A1 (BML-CM110-0100) from Enzo.

## Cells and cell culture

MEF (kind gift of Noboru Mizushima, Tokyo Medical and Dental University, Tokyo, Japan), HeLa (WT, FIP200 KO, ATG7 KO, ATG9 KO, ATG16L1 KO; Nakamura et al., 2020), and Cos7 (ATCC) cells were cultured in Dulbecco's modified Eagle medium (DMEM; Sigma Aldrich) supplemented with 10% fetal bovine serum (FBS), 1% penicillin/streptomycin, and non-essential amino acids at 37°C with 5% $CO_2$. Cells were routinely tested for mycoplasma contamination using the LookOut mycoplasma PCR detection kit (Sigma Aldrich).

## Transfection

Transfection of DNA constructs was performed using X-tremeGENE HP transfection reagent (Roche) according to the manufacturer's directions.

## Cholesterol staining

Free cholesterol was visualized using Filipin (Sigma-Aldrich) staining. Cells were fixed in 3% paraformaldehyde in PBS for 1 h at r.t., washed three times with PBS containing 1.5 mg/ml glycine, and incubated with 0.05 mg/ml Filipin in PBS containing 10% FBS for 30 min at 37°C, then an additional 1–2 h at r.t.

## Immunoblotting

Cells were lysed in cold lysis buffer (20 mM Tris-HCl pH 8, 300 mM KCl, 10% Glycerol, 0.25% Nonidet P-40, 0.5 mM EDTA, 1 mM PMSF, 1× complete protease inhibitor [Roche]) and cleared by centrifugation (20 min/18,213 × $g$/4°C). Protein concentrations were determined through Bradford assay (Bio-Rad Protein Reagent) and lysates were normalized. Proteins were separated by SDS-PAGE and transferred to a 0.2-µm nitrocellulose membrane (Bio-Rad) using a Trans-Blot Turbo transfer system (Bio-Rad). Membranes were blocked using 5% skim milk in TBST and incubated with primary antibody overnight at 4°C. Protein detection was carried out using chemiluminescence (Bio-Rad) and imaged using a ChemiDoc imaging system (Bio-Rad).

## Immunofluorescence

Cells were grown on no. 1.5 glass coverslips and fixed in 4% paraformaldehyde in PBS for 10 min at r.t. Cells were washed three times with PBS containing 1.5 mg/ml glycine, permeabilized in 0.25% Triton X-100 in PBS for 5 min and washed three times with PBS. Cells were blocked with 5% donkey serum for 30 min followed by a 1–2 h incubation with primary antibody at r.t. Cells were washed three times with PBS and incubated with Alexa Fluor-conjugated secondary antibodies for 30 min at r.t. Cells were washed three times with PBS and mounted on slides using ProLong Diamond antifade mountant (Thermo Fisher Scientific). 4′,6-diamidino-2-phenylindole (DAPI) was added to the second PBS wash (1 µg/ml) to stain nuclei.

Immunofluorescence imaging was performed on a Leica SP8 FALCON inverted confocal system (Leica Microsystems) equipped with a HC PL APO 63×/1.40 oil immersion lens and a temperature-controlled hood maintained at 37°C and 5% $CO_2$. The system was controlled by Leica Application Suite X (LAS X) acquisition software. DAPI was excited using a 405 nm Diode laser, and FITC/Alexa488, mCherry/Alexa568, and Alexa647 were excited using a tuned white light laser. Scanning was performed in line-by-line sequential mode.

Fluorescence intensity quantification and image processing was performed using ImageJ—FIJI distribution (National Institutes of Health; Schindelin et al., 2012).

## Microscale thermophoresis (MST)

The 22-NBD cholesterol (22-[N-[7-Nitrobenz-2-oxa-1,3-diazol-4-yl] amino]-23,24-bisnor-5-cholen-3-ol, [3,20S]-20-Methyl-21-[7-nitro-4-benzofurazanyl]-amino]-pregn-5-en-3-ol; N2161; Sigma-Aldrich) and N-methyl NBD (ENA370563614; Sigma-Aldrich) were used for binding affinity measurement of MakA proteins using a Nanotemper Monolith NT.115 instrument. N-methyl NBD or 22-NBD cholesterol (final concentration as 80 nM) was mixed with different concentrations of protein in MST buffer (25 mM Bis-tris, 109.5 mM NaCl, 5.4 mM KCl, 0.4 mM $MgSO_4$, 0.45 mM $CaCl_2$, 0.05% Tween-20 v/v) at the indicated pH. Experiments were performed at 22°C, and samples were filled into standard capillaries for measurement.

## QCM-D

The QCM-D measurement was performed on silica ($SiO_2$)-coated sensors using an Advanced Multichannel System (AWS X4). Sensors were cleaned with ozone prior to use. Supported lipid bilayers were formed using extruded PM mix liposomes (50 nm, 0.1 mg/ml) in QCM-D buffer (20 mM citric acid, 50 mM KCl, 0.1 mM EDTA, pH 4.5) at 37°C. After forming supported lipid bilayers, sensors were rinsed with liposome buffer at according pH, which was the buffer for protein adsorption. Wild-type MakA (final concentration as 10 µM) was injected into chamber, and the bilayer was incubated at 25°C until steady condition before rinsing with liposome buffer at pH 7.6. The frequency shift ($\Delta F$) of the third overtone was converted into a mass value through Sauerbrey equation.

## Reconstitution of MakA in nanodiscs

Membrane Scaffold Protein (MSP1E3D1) was used for nanodisc preparation. The stock solution of *E. coli* BL21 (DE3) Gold transfected with MSP1E3D1 plasmid was a kind gift from Linda Sandblad (Umeå university, Umeå, Sweden). The expression and purification of MSP1E3D1 protein were similar to the procedure of MakA proteins with few adjustments. Briefly, MSP1E3D1 protein was expressed in Terrific Broth (TB broth). Induction was initiated at $OD_{600}$ (1.0) with 1 mM IPTG followed by 4 h growth at 37°C. Purification was comprised with three steps: his-tag affinity purification with 5 ml HisTrap HP column, TEV cleavage overnight through dialysis followed by another his-tag affinity purification and then size exclusion chromatography with buffer (20 mM tris, 100 mM NaCl, 0.5 mM EDTA, pH 7.4). Fractions of interest were concentrated and supplemented with 1 mM $NaN_3$. Protein was then aliquoted and flash-frozen with liquid nitrogen.

For preparation of empty nanodiscs, lipid mix (DOPC/DOPE/cholesterol = 3/1/1 in molar ratio) was rehydrated in ND buffer (20 mM tris, 100 mM NaCl, pH 7.4) containing 300 mM sodium cholate to dissolve lipids into 50 mM stock. The lipid mixture (5.6 mM in final) and MSP1E3D1 (112 µM in final) were incubated for 90 min at 4°C. The mixture was then dialyzed overnight against ND buffer containing Biobeads and then loaded on a Superdex 200 increase 10/300GL column, which was equilibrated with ND buffer. Fractions of interest were concentrated to ~1 mg/ml and frozen at –80°C until further use.

For constitution of MakA into nanodiscs, empty nanodisc was buffer-exchanged to liposome buffer at pH 5.4 through NAP-5 column. Wild-type MakA was then added. MakA, MSP1E3D1, and lipid mix were used in a ratio of 1 (1 nmol):1 (1 nmol):50 (50 nmol). After incubation at r.t. for 150 min to induce pH-dependent MakA-cholesterol interaction on nanodiscs, the mixture was loaded on a Superdex 200 increase 10/300GL column to separate aggregates and/or empty nanodiscs from nanodiscs with reconstituted MakA protein.

### Negative staining and CryoEM
For negative staining, 4 µl of nanodiscs reconstituted with MakA (4 µM) was applied on a glow-discharged copper grid with thin pure carbon film (Ted Pella; 300 mesh). After incubation for 1 min, the sample was blotted with filter paper, washed with water, and then stained with 1.5% uranyl acetate. The grids can be stored at r.t. for a few months or longer under dry condition. The images were recorded manually with a FEI Talos L 120C TEM, operating at an accelerating voltage of 120 kV, electron source as LaB6 filament, and Ceta 4k × 4k CMOS detector.

For CryoEM, 4 µl of nanodiscs reconstituted with MakA (12 µM) was applied to a glow-discharged copper grid with 2-nm continuous carbon film (Quantifoil, QF 2/1, 300 mesh), blotted for 5 s with blot force as –5 in 100% humidity at 4°C, and flash-frozen in liquid ethane with a FEI Vitrobot plunge freezer. The grid was imaged using a FEI Titan Krios TEM operating at an accelerating voltage of 300 kV equipped with a Gatan Bio-Quantum energy filter and a K2 direct detector with C2 aperture of 70 µm and objective aperture of 100 µm. 4,500 micrographs were acquired using EPU software with following settings: nominal magnification of 165,000× (0.82 Å/pixel), defocus range –0.6 to –1.8 µm, total dose of 45.6 e/Å$^2$, and total exposure time of 4 s.

For CryoEM data processing, 4,500 micrographs were processed using CryoSPARC (Punjani et al., 2017), with patch-based motion correction, patch CTF estimation, manually picking particles as template for automatic picking and 2D classification. After six times of 2D classification, 20 classes of particles were achieved with overall resolution of ~20 Å.

### Fluorescent labeling of MakA proteins
Alexa Fluor 568 (AF568) labeling of monomeric MakA, oligomeric MakA, and MakA mutant proteins was performed using an Alexa Fluor 568 protein labeling kit (Cat. no: A20003; Thermo Fisher Scientific) according to the manufacturer's instruction. Briefly, the target protein (5.5 mg/ml, 140 µM) was incubated with 1.5 equivalents Alexa Fluor 568 (210 µM) in

0.2 M sodium bicarbonate buffer pH 8.3 for 2 h at r.t. For the FITC labeling, MakA protein (100 µM) was incubated with 10 equivalents FITC (1,000 µM) in 50 mM sodium phosphate buffer at pH 8.0 with 150 mM NaCl overnight at r.t. Each labeled protein was separated from free fluorescent dye using a gel filtration column (GE Healthcare illustra NAP-10 Columns Sephadex G-25), followed by ESI-MS analysis (Agilent 6230 TOF LC/MS) and SDS-PAGE analysis. The degree of labeling (DOL) is calculated from following equations, $CF = A280$ free dye/Amax free dye, $DOL = Amax × ε280/([A280–Amax × CF] × ε578)$. A280 is the absorbance of the protein–dye conjugate at 280 nm; A578 Amax is the absorbance of the protein–dye conjugate at its maximum absorbance; ε280 is the extinction coefficient of the protein at 280 nm in cm$^{-1}$ M$^{-1}$; εdye is the extinction coefficient of the dye at its maximum absorbance in cm$^{-1}$ M$^{-1}$; and CF is the correction factor and CF values for Alexa Fluor 568 are 0.46, 0.3 for FITC. The DOLs of AF568-labeled MakA_M, MakA_O, MakA$^{I236D\&I237D}$, and MakA$^{W243D}$ are 0.54, 0.63, 0.49, and 0.56, respectively. The DOL of FITC-labeled monomeric MakA is 0.52.

### Statistical analysis
All data are shown as mean ± SD. Statistical significance was determined by one-way ANOVA or by Student's $t$ test (two-tailed, unpaired), as indicated in the corresponding figure legend using GraphPad Prism v.8. P values are indicated in the corresponding figure legend.

### Online supplemental material
Fig. S1 shows MakA's interaction with different lipids on liposomes. Fig. S2 shows the kinetics and affinity of MakA's interaction with the membrane. Fig. S3 shows a structure-function analysis of MakA-Cholesterol interaction on the membrane. Fig. S4 shows the gel-filtration plot of MakA and mutants. Fig. S5 shows supplemental immunofluorescence data relating to MakA-induced membrane damage and non-canonical autophagy induction.

## Acknowledgments
We would like to thank Karina Persson for providing MakB and MakE proteins, Pravin Kumar for help with liposome assay and GUV preparation, Marta Bally and Hudson Pace for the help with QCM-D assay, Michael Hall for help with sample preparation and data acquisition in TEM and CryoEM, and Karim Rafie for constructive suggestions in CryoEM data processing.

We acknowledge the Biochemical Imaging Center at Umeå University and the National Microscopy Infrastructure, NMI (VR-RFI 2019-00217) for providing assistance in microscopy. TEM and CryoEM were collected at the Umeå Centre for Electron Microscopy (UCEM), a SciLifeLab National Cryo-EM facility and part of National Microscopy Infrastructure, NMI (VR-RFI 2016-00968), supported by instrumentation grants from the Knut and Alice Wallenberg Foundation and the Kempe Foundation.

This work was supported by the European Research Council (ChemBioAP), Vetenskapsrådet (Nr. 2018-04585), the Knut and Alice Wallenberg Foundation and the Göran Gustafsson

Foundation for Research in Natural Sciences and Medicine to Y.-W. Wu.

The authors declare no competing financial interests.

Author contributions: X. Jia, D.P. Corkery, A. Knyazeva, and Y. Wu designed the research; X. Jia, D.P. Corkery, A. Knyazeva, Y. Zhang, and S. Castro-Gonzalez performed the research; X. Jia, D.P. Corkery, A. Knyazeva, L.-A. Carlsson, and Y. Wu analyzed the data; S. Nakamura and T. Yoshimori contributed reagents; X. Jia, D.P. Corkery, and Y. Wu wrote the paper; and Y. Wu conceptualized and supervised the project and acquired funding.

Submitted: 9 June 2022

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

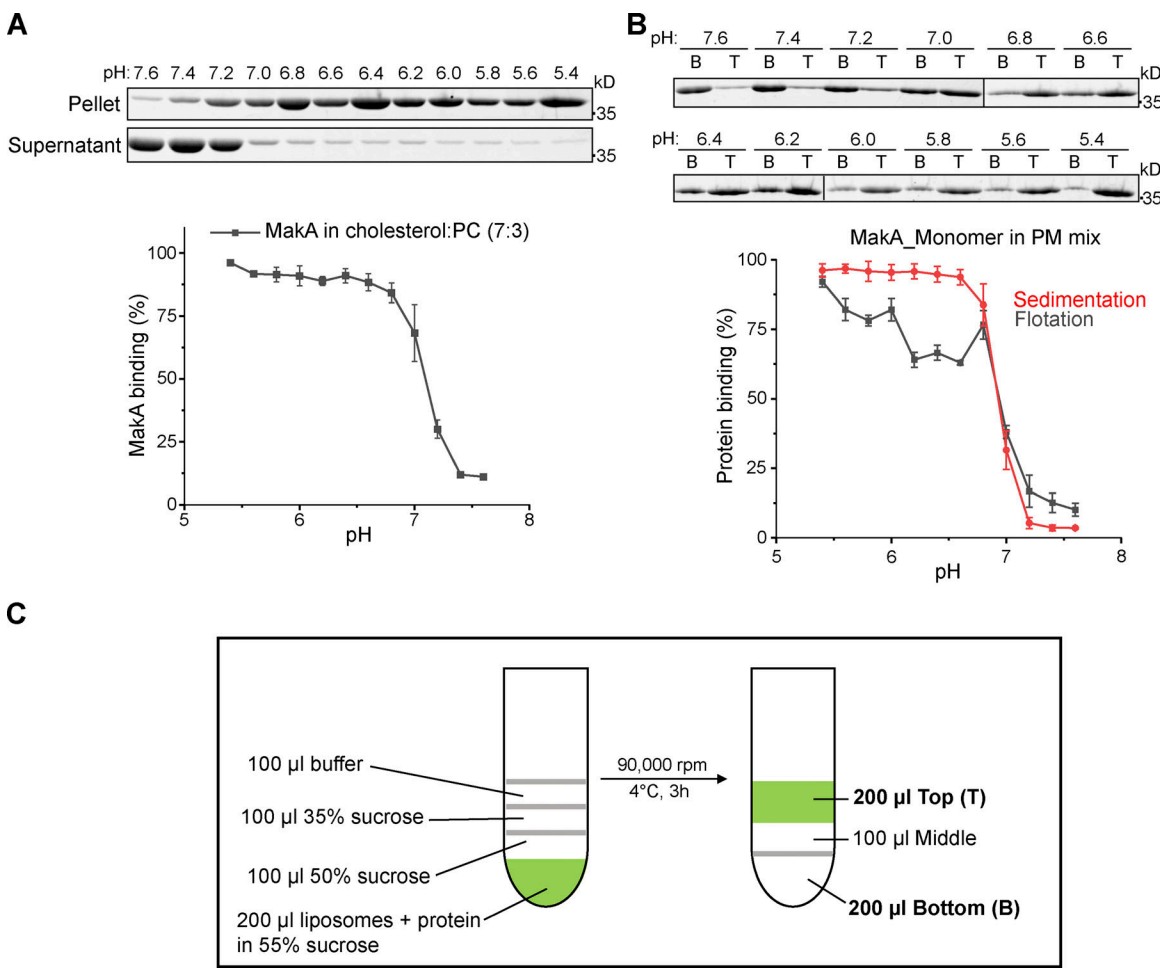

Figure S1. **MakA's interaction with different lipids on liposomes.** Wild-type monomeric MakA was used in the following assays. **(A)** SDS-PAGE and analysis of MakA's liposome sedimentation assay from pH 5.4–7.6; the liposomes contain 70% cholesterol and 30% DOPC. Data are shown as mean ± SD from three independent experiments. **(B)** SDS-PAGE and analysis of MakA's liposome flotation assay from pH 5.4–7.6; the liposomes are PM mix (50% DOPC, 20% cholesterol, 10% DOPE, 5% DOPS, 5% Soy PI, 10% Brain SM). B: bottom fraction; T: top fraction. Data are shown as mean ± SD from three independent experiments. **(C)** Illustration of liposome flotation assay. Liposomes and protein mix in 55% sucrose layered with 50% sucrose, 35% sucrose and 0% sucrose (buffer). Green color indicates NBD--PE. Source data are available for this figure: SourceData FS1.

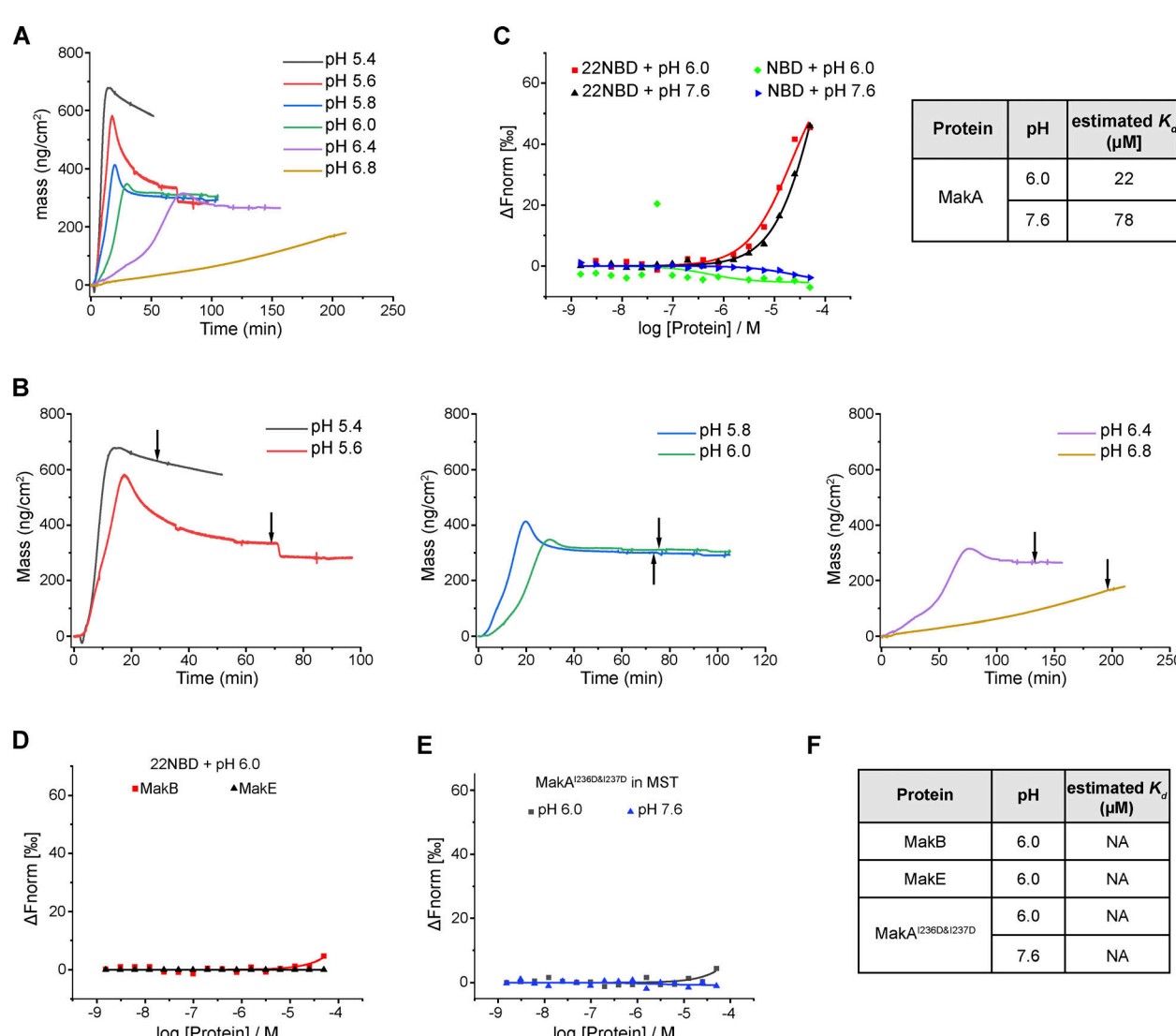

Figure S2. **Kinetics and affinity of MakA's interaction with the membrane.** Wild-type monomeric MakA was used in the following assays. **(A)** QCM-D measurement of MakA (10 µM) binding to supported lipid bilayers (50% DOPC, 20% cholesterol, 10% DOPE, 5% DOPS, 5% Soy PI, 10% Brain SM) from pH 5.4–6.8. The frequency shift ($\Delta F$) of the third overtone was converted into a mass value via Sauerbrey equation. 0 min represents the time of MakA injection. **(B)** QCM-D measurement of MakA (10 µM) binding to supported lipid bilayers. Arrows indicates time to rinse with 25 mM Bis-Tris (pH 7.6), 109.5 mM NaCl, 5.4 mM KCl, 0.4 mM MgSO4, 0.45 mM CaCl2. **(C)** MST analysis of MakA's interaction with cholesterol in solution. 22NBD:22-NBD cholesterol; NBD: N-methyl NBD; MakA (50 µM to 1.5 nM) were mixed with 22NBD (80 nM) or NBD (80 nM) at pH 6.0 or 7.6. $K_d$ values were determined by the Nanotemper Monolith MO.Affinity software. **(D)** MST analysis of MakB's or MakE's interaction with cholesterol in solution. 6xHis-MakB (50 µM to 1.5 nM) or MakE (50 µM to 1.5 nM) were mixed with 22-NBD cholesterol (80 nM) at pH 6.0; **(E)** MST analysis of the interaction of MakA[I236D&I237D] with cholesterol in solution. MakA[I236D&I237D] (50 µM to 1.5 nM) were mixed with 22-NBD cholesterol (80 nM) at pH 6.0 or 7.6. **(F)** $K_d$ of measurements in D and E. NA, not available.

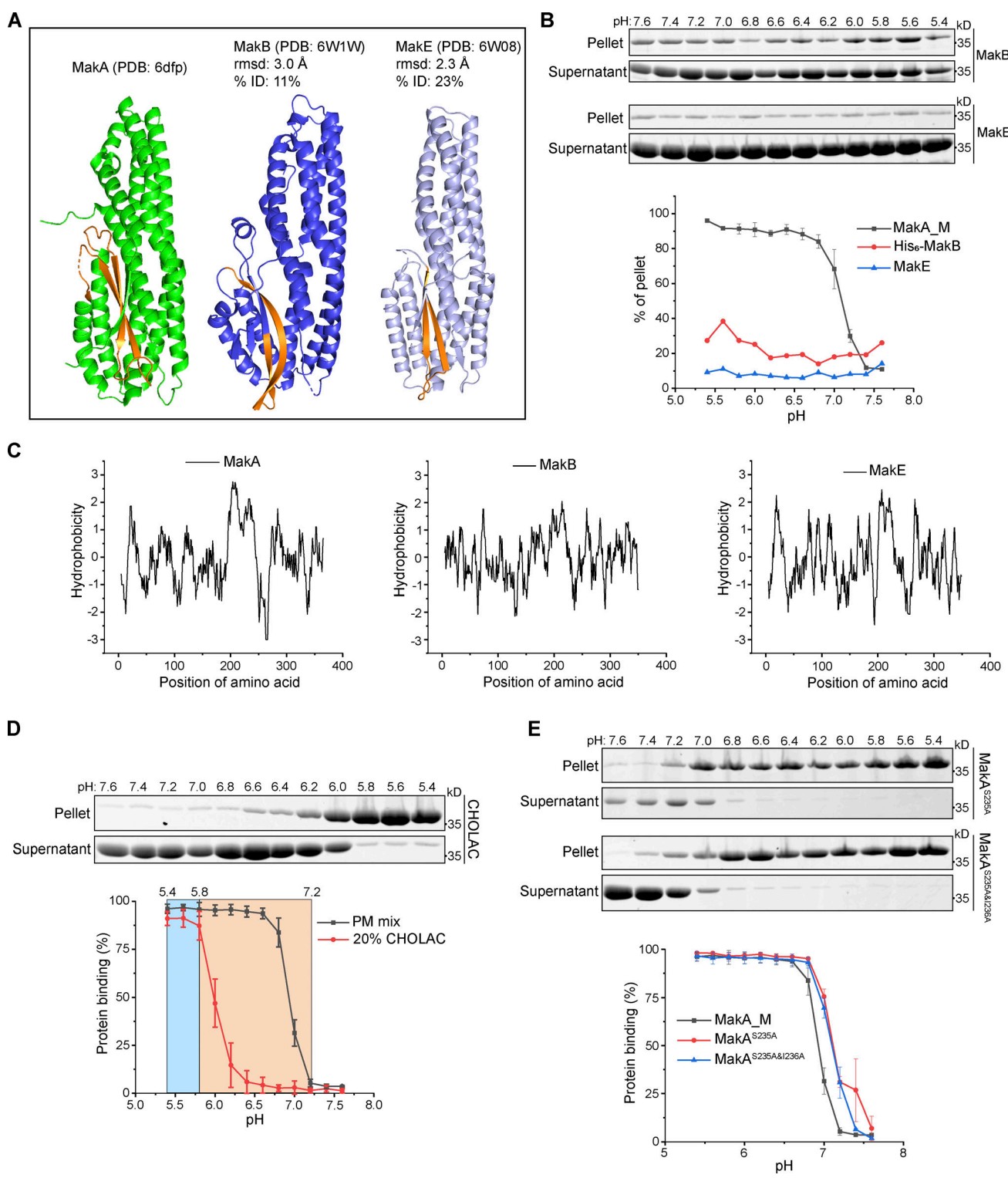

Figure S3. **Structure-function analysis of MakA-Cholesterol interaction on the membrane. (A)** Structure comparison of MakA, MakB, and MakE via DALI. Hydrophobic regions are highlighted with orange. **(B)** SDS-PAGE and analysis of MakB's or MakE's liposome sedimentation assay from pH 5.4–7.6. The liposomes contained 70% cholesterol and 30% DOPC. Data are shown as mean ± SD from three independent experiments. **(C)** Kyte & Doolittle hydrophobicity plot of MakA, MakB, and MakE created from ProtScale. **(D)** SDS-PAGE and analysis of MakA's liposome sedimentation assay from pH 5.4–7.6, The liposomes contained 80% DOPC and 20% cholesterol acetate or PM mix (50% DOPC, 20% cholesterol, 10% DOPE, 5% DOPS, 5% Soy PI, 10% Brain SM). CHOLAC: cholesterol acetate. Data are shown as mean ± SD from three independent experiments. **(E)** SDS-PAGE and analysis of liposome sedimentation assay of MakA[S235A] and MakA[S235A&I236A] from pH 5.4–7.6. Liposomes are PM mix (50% DOPC, 20% cholesterol, 10% DOPE, 5% DOPS, 5% Soy PI, 10% Brain SM). Data are shown as mean ± SD from three independent experiments. Source data are available for this figure: SourceData FS3.

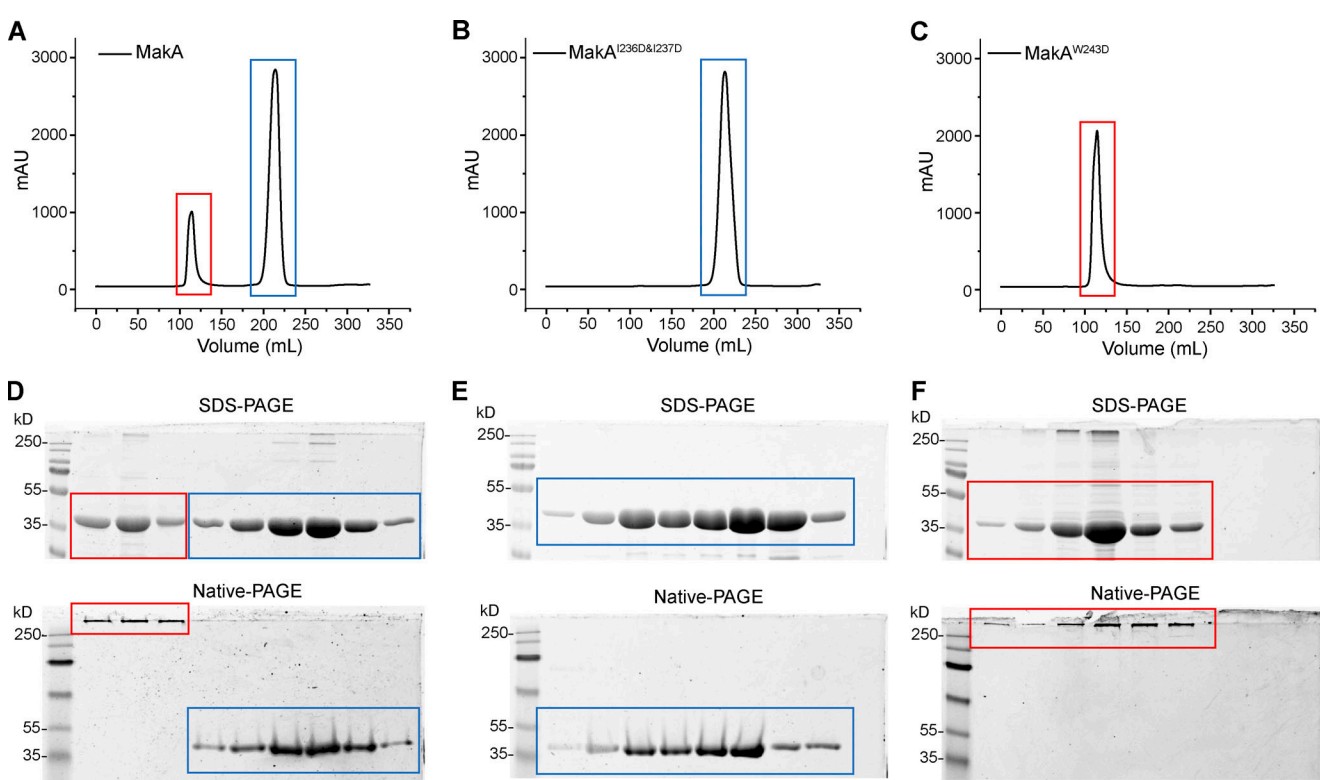

Figure S4. **Gel-filtration plot of MakA and mutants.** Peak at ~110 ml represents oligomer (red box), and peak at ~220 ml represents monomer (blue box). **(A)** Gel-filtration chromatography of wild-type MakA. **(B)** Gel-filtration chromatography of MakA^I236D&I237D. **(C)** Gel-filtration chromatography of MakA^W243D. **(D)** SDS-PAGE and Native-PAGE analysis of fractions in A. **(E)** SDS-PAGE and Native-PAGE analysis of fractions in B. **(F)** SDS-PAGE and Native-PAGE analysis of fractions in C. Source data are available for this figure: SourceData FS4.

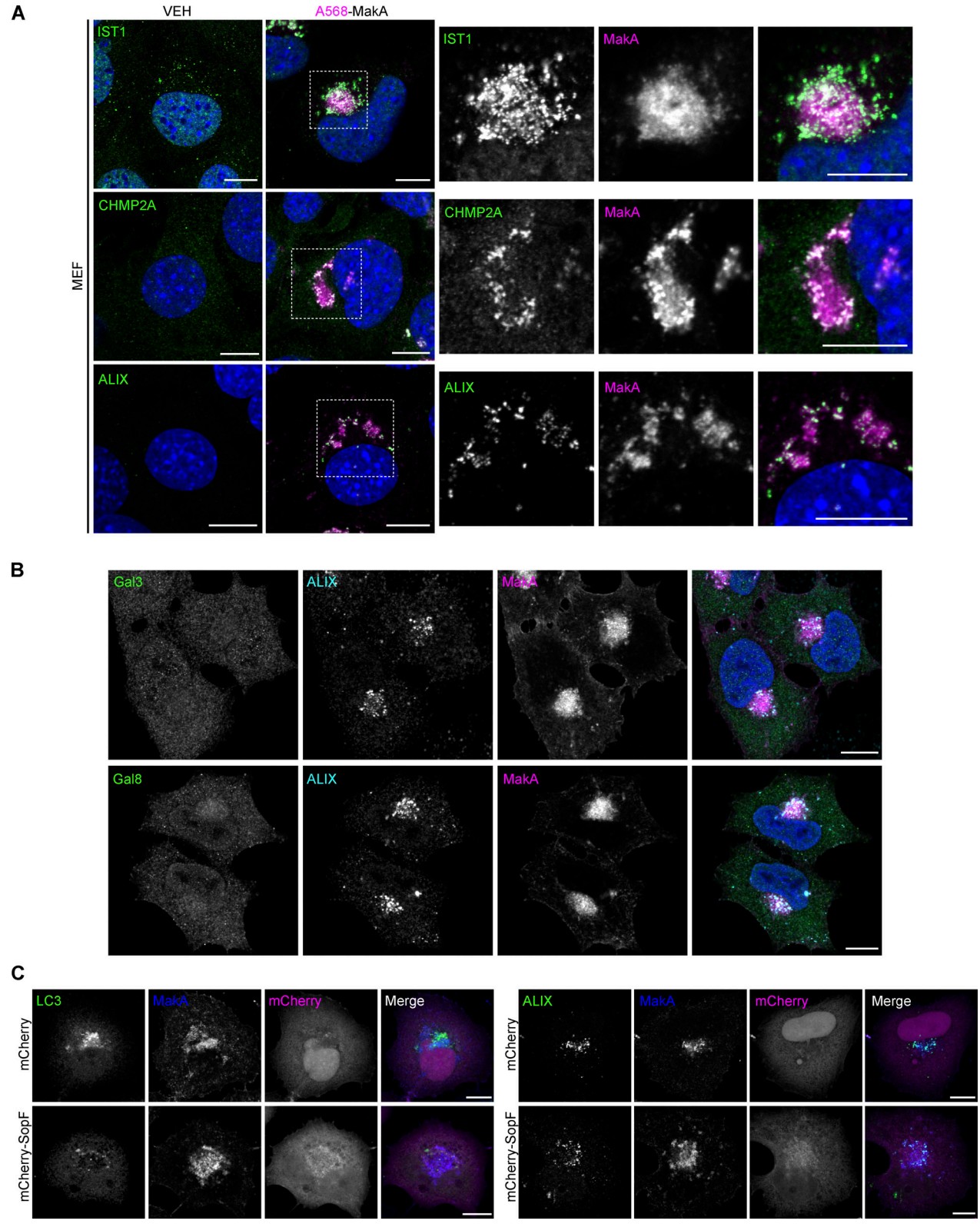

Figure S5. **Low pH triggers MakA-induced membrane damage and non-canonical autophagy—Supplemental immunofluorescence data. (A)** Representative confocal images of MEF cells treated with 250 nM Alexa568-MakA for 16 h and co-stained for endogenous IST1, CHMP2A, or ALIX, as indicated. Nuclei were counterstained with DAPI. Scale bars, 10 μm. Alexa568-MakA refers to Alexa568 labeled monomeric MakA. **(B)** Representative confocal images of HeLa cells treated with 125 nM Alexa568-MakA for 16 h and co-stained for endogenous ALIX and Gal3 or Gal8, as indicated. Nuclei were counterstained with DAPI. Scale bars, 10 μm. Alexa568-MakA refers to Alexa568 labeled monomeric MakA. **(C)** Representative confocal images of HeLa cells transfected with mCherry or mCherry-SopF and treated with 250 nM FITC-MakA for 16 h. Cells were co-stained for endogenous LC3 (left) or ALIX (right). Scale bars, 10 μm. FITC-MakA refers to FITC labeled monomeric MakA.

