## [Peer Review File · The Journal of Cell Biology]

V. Cholerae MakA is a cholesterol-binding pore-forming toxin that induces non-canonical autophagy

Xiaotong Jia, Anastasia Knyazeva, Yu Zhang, Sergio Castro-Gonzalez, Shuhei Nakamura, Lars-Anders Carlson, Tamotsu Yoshimori, Dale Corkery, and Yaowen Wu

Corresponding Author(s): Yaowen Wu, Umeå University and Dale Corkery, Umeå University

Review Timeline:

Submission Date:	2022-06-09
Editorial Decision:	2022-07-14
Revision Received:	2022-08-16
Editorial Decision:	2022-09-13
Revision Received:	2022-09-14

Monitoring Editor: Craig Roy

Scientific Editor: Tim Fessenden

Transaction Report:

DOI: <https://doi.org/10.1083/jcb.202206040>

July 14, 2022

Re: JCB manuscript #202206040

Prof. Yaowen Wu
Umeå University
Department of Chemistry
Linnaeus väg 6
Umeå 90187
Sweden

Dear Prof. Wu,

Thank you for submitting your manuscript entitled "The Vibrio Cholerae cytotoxic MakA is a cholesterol-binding toxin that induces non-canonical autophagy via pH-dependent pore-formation". Your manuscript has been assessed by expert reviewers, whose comments are appended below. Although the reviewers express potential interest in this work, significant concerns unfortunately preclude publication of the current version of the manuscript in JCB.

You will see that, while reviewers appreciated the conceptual advance showing a dependence of MakA on cholesterol by this work, they found it did not make full use of existing literature and structural details reported by others. In addition, Reviewer 2 sought clarification on whether MakA can use cholesterol to enter cells at the surface, or stringently required low pH, towards a better understanding of this protein's use of cholesterol as a receptor. We invite a revision addressing all issues raised by all reviewers.

Please let us know if you are able to address the major issues outlined above and wish to submit a revised manuscript to JCB. Note that a substantial amount of additional experimental data likely would be needed to satisfactorily address the concerns of the reviewers. The typical timeframe for revisions is three to four months. While most universities and institutes have reopened labs and allowed researchers to begin working at nearly pre-pandemic levels, we at JCB realize that the lingering effects of the COVID-19 pandemic may still be impacting some aspects of your work, including the acquisition of equipment and reagents. Therefore, if you anticipate any difficulties in meeting this aforementioned revision time limit, please contact us and we can work with you to find an appropriate time frame for resubmission. Please note that papers are generally considered through only one revision cycle, so any revised manuscript will likely be either accepted or rejected.

If you choose to revise and resubmit your manuscript, please also attend to the following editorial points. Please direct any editorial questions to the journal office.

GENERAL GUIDELINES:

Text limits: Character count is < 40,000, not including spaces. Count includes title page, abstract, introduction, results, discussion, and acknowledgments. Count does not include materials and methods, figure legends, references, tables, or supplemental legends.

Figures: Your manuscript may have up to 10 main text figures. To avoid delays in production, figures must be prepared according to the policies outlined in our Instructions to Authors, under Data Presentation, <https://jcb.rupress.org/site/misc/ifora.xhtml>. All figures in accepted manuscripts will be screened prior to publication.

Supplemental information: There are strict limits on the allowable amount of supplemental data. Your manuscript may have up to 5 supplemental figures. Up to 10 supplemental videos or flash animations are allowed. A summary of all supplemental material should appear at the end of the Materials and methods section.

Please note that JCB now requires authors to submit Source Data used to generate figures containing gels and Western blots with all revised manuscripts. This Source Data consists of fully uncropped and unprocessed images for each gel/blot displayed in the main and supplemental figures. Since your paper includes cropped gel and/or blot images, please be sure to provide one Source Data file for each figure that contains gels and/or blots along with your revised manuscript files. File names for Source Data figures should be alphanumeric without any spaces or special characters (i.e., SourceDataF#, where F# refers to the associated main figure number or SourceDataFS# for those associated with Supplementary figures). The lanes of the gels/blots should be labeled as they are in the associated figure, the place where cropping was applied should be marked (with a box), and molecular weight/size standards should be labeled wherever possible.

If you choose to resubmit, please include a cover letter addressing the reviewers' comments point by point. Please also highlight all changes in the text of the manuscript.

Regardless of how you choose to proceed, we hope that the comments below will prove constructive as your work progresses. We would be happy to discuss them further once you've had a chance to consider the points raised. You can contact the journal office with any questions, cellbio@rockefeller.edu or call (212) 327-8588.

Thank you for thinking of JCB as an appropriate place to publish your work.

Sincerely,

Craig Roy
Monitoring Editor
Journal of Cell Biology

Tim Fessenden
Scientific Editor
Journal of Cell Biology

Reviewer #1 (Comments to the Authors (Required)):

In the manuscript, Jia et al. report MakA as a new pH-dependent cholesterol-binding pore-forming toxin. Interaction between MakA and cholesterol were confirmed by a few methods. The binding was obvious below pH 6.8. MaKA is structurally similar to ClyA, but the binding with cholesterol is pH dependent. Residues I236 and I237 were found be critical for interaction with cholesterol, different from cholesterol-dependent cytolysins. MakA forms pores on membranes with cholesterol at pH 5.4 but not pH 6.8, indicating complicated regulation and multiple states of MakA. MakA in host cells will form pores at low pH and cause membrane damage that recruits ESCRT machinery and non-canonical autophagy. These findings reveal the molecular mechanisms of MakA regulation of host signaling pathways. The manuscript is well prepared but the following issues need to be addressed:

Major points:

1. Introduction, Paragraph 5: "... and subsequent inhibition of canonical and non-canonical autophagic responses.", "inhibition" is causing confusion and needs to be explained as MakA is inducing non-canonical autophagy throughout the manuscript.
2. Fig. 3A: Did the cryo-EM data result in structure models (high-res or not)? This needs to be discussed and explained. The structure information will be helpful and strengthen the manuscript. Please estimate the number of monomers in a pore. How many monomers are taken up in an endosome? Will the apparent concentrations be high enough? μM ?
3. Low pH triggers MakA-induced membrane damage and non-canonical autophagy, Paragraph 3: "Treatment with both BafA1 and MakA results in LC3 II accumulation similar to BafA1 alone, suggesting inhibition of MakA-dependent LC3 lipidation." This is only one possibility though the text below shows that it may be the situation. Other possibilities are inhibition of BafA1-induced lipidation or LC3II is consumed out. This sentence needs to be revised to fix the logical problem.
4. Discussion, Last Paragraph: "tubular structure of MakA at pH 6.5" should be discussed in detail as it is related to the findings described in the manuscript. Did the authors observe any similar structures? Any indication for function or regulation? Top views of 2D averages seem to be similar with tubular structure top view, do the authors have any idea about the aggregate structure? Could it be tubular?
5. The similarity between MakA and ClyA should be explained with more details to illustrate possible mechanisms for cholesterol-binding and pH regulation. A structural comparison with cholesterol-dependent cytolysins will also provide important information.

Minor points:

1. Introduction, Paragraph 3: "... in response to activation of several non-autophagic pathways (11) ...", "non-autophagic" should be "non-canonical".
2. Fig. S3E: "S235S" should be S235A.
3. For clarity change

Importantly, co-treatment with MakA and BafA1 completely blocks MakA-induced LC3 lipidation (Fig. 5A, 5B).

To

Importantly, co-treatment with MakA and BafA1 in cells deficient for basal autophagy completely blocks MakA-induced LC3 lipidation (Fig. 5A, 5B).

Reviewer #2 (Comments to the Authors (Required)):

This paper finds that the Pore forming toxin MakA interacts with Cholesterol and forms oligomers at low pH. Another major finding is a mutant that locks the toxin in an oligomeric state. The results are of interest and generally well conducted. The work expands and work by others with interesting new findings. The work itself is solid. The primary issues with the paper are the overinterpretation of some results and some conclusions not fully supported and lack of consideration of results in context of published literature. A redrafted manuscript (especially discussion) should address all of the following major and minor comments.

Major comments

- 1) The paper discusses multiple cholesterol dependent cytolysins (CDC) without mentioning the pivotal work by Michael Jennings in 2014 (PNAS <https://doi.org/10.1073/pnas.1412703111>) and 2020 (Science Advances DOI: 10.1126/sciadv.aaz4926). These studies reveal CDCs use glycans as cellular receptors which drives their cell tropism, in contrast to prior considerations that they bind cholesterol directly in membranes. This is important here as a recent 2022 paper by Herrera et al showed that while MakA is cytotoxic at high concentrations on many different cells, it shows cell tropism similar to many of the CDCs, suggesting it does use a receptor. Further, it was functional at nanomolar concentrations when combined with MakB and MakE, consistent with high affinity engagement. The finding that MakA CAN bind cholesterol and may even depend on cholesterol and pH for insertion and/or oligomerization are well done and move the field forward in our understanding of MakA. However, the data in this paper do not support that MakA is a receptor. Most importantly, the binding affinity is in the micromolar ranges and MakE and MakB were found not to bind cholesterol at all. It is suggested the authors review this body of literature and update the manuscript to accurately reflect the role of cholesterol in CDCs and remove all comments that cholesterol may be a receptor for MakA. The discussion should be tempered that cholesterol is "important" for MakA interactions and possibly insertion/oligomerization when functioning as a single toxin, which may be different that the role of cholesterol when MakA is part of the high affinity tripartite toxin that has specific cell tropism.
- 2) Although there is no overlap in named authors, scientists from the same institution previously reported that this toxin binds to phosphatitic acid. Whereas Fig 1B shows no interaction. Discussion should address the difference in the two studies and support why this study should take precedence? Perhaps technical?
- 3) As part of point 1, Fig. 1 and S1 nicely show that MakA interactions with cholesterol loaded liposomes by two methods. However, it is important they note that, cholesterol CAN interact with MakA in vitro at a pH ideal for MakA-induced LC3 lipidation process. This does not necessarily lead to their conclusion that MakA will bind to membrane cholesterol in cells. Further, when the toxin would be in this environment is not clear? The model seems to suggest interaction prior to endocytosis with assembly occurring at pH6.5 in the acidified vacuole. This does not support a role of cholesterol as a receptor for MakA, as this should occur at the plasma membrane surface, well before acid conditions encountered. The model should be amended to accurately consider When and Where MakA would encounter pH and cholesterol as imaged in Fig S8 at the plasma membrane surface is not likely the location
- 4) The results section regarding Fig 5 are very complicated and the text is difficult to follow making it hard to assess the results regarding non-canonical autophagy although the data seem to support the conclusions drawn.
- 5) The discussion is rather abrupt and does not integrate well with the literature in this field and the model as presented in Figure S8 is not well supported.

Minor points

- 1) *Vibrio cholerae* and *V. cholerae* are spelled wrong throughout the paper.

- 2) Figure 1E Seems to present only one image. Are these cells fixed? is the toxin on the surface or inside the cell? Cannot tell from these images. Quantification shows only a single cell.
- 3) Please indicate in legend that representative images are representative of How many (n) cells and experiments?? Reproducibility information on microscopy is missing from the legends.
- 4) Crystal structures are all now published. Please cite all references when using the PDB ID#. See papers by Herrera et al Infect Immun 2022 and by Nadeem et al PNAS 2021.
- 5) Papers by others support other models of death than non-canonical autophagy. How do results here compare with recent papers on MakA? How would the model account for evidence this works at much lower concentration in concert with MakB and MakE in specific cell lines

Reviewer #3 (Comments to the Authors (Required)):

Summary

The authors characterize binding of a pore forming toxin MakA to cholesterol. They identify key residues for binding, and show that binding requires an acidic pH. Using CryoEM they show that MakA can form 3nm pores, in at least the conditions they used. In prior studies they showed that MakA induces formation of endosomal aggregates positive for autophagic markers (e.g. LC3). Here they show that MakA induces these cellular aggregates through generating small pores, and independently of the autophagic response. Overall, the experiments are well designed and the data are of high quality. The paper is of interest to microbiologists since pore forming toxins are major bacterial virulence factors, but also to cell biologists since it examines how endosomes/lysosomes can be repaired. The main weakness of the paper is they do not take advantage of prior structural data available for MakA.

Major comments

-An important consideration is the nature of MakA protein monomers. How stable is MakA protein at 37 degrees at different pH, independently of any lipids? Does it lose membrane binding/perforating activity due to loss of stability of the protein at certain pH?

-The cryo EM images in Figure 3A are impressive. Can a combined model be generated? Also, is it expected that these maximally 3nm pores, or are these images representing an intermediate in the assembly of larger pores? How do these images compare to other PFTs?

-pg8, last paragraph of Discussion. The authors acknowledge that MakA was shown in another paper (ref 49) to oligomerize into tubular structures. That paper has important structural data that should be leveraged as much as possible to provide insight into the data acquired here. Importantly, are the authors visualizing an alternate form of the toxin? Or do they think the big tube-like oligomers are forming within endosomes after acidification?

Minor comments:

-page and line numbers would help this reviewer

-Should MakA be classified as a pore-forming toxin or a cholesterol-dependent cytolysin? What is the consensus for this?

-pg2, second paragraph "pores range from a few nm to 50 nm". This is not entirely accurate... monomers can affect membranes with channel like activity, and form other structures such as arcs, but they are not really pores at that point (in the way that we typically think about them). Some of the atomic force microscopy studies of other toxins should be cited and discussed here.

-pg3, please define "PM mix liposomes"?

-pg7, "In contrast, MakA pore on the membrane (~ 3 nm) is smaller, which is doable for the passage of ions or small molecules but not proteins". I'm not sure if this is true, since the type 3 secretion system from bacteria has a narrow shaft and yet translocates proteins into host cells. Also, the word "doable" is a slang term.

Dear Editor,

We appreciate that the reviewers' critical comments help improve our manuscript. We have performed new experiments and analysis to address reviewers' questions. These new results are highlighted in the manuscript. We provide point-by-point response to reviewers' comments as shown below.

Best regards,
Yaowen Wu

Point-by-point responses to Reviewers:

Reviewer #1 (Comments to the Authors (Required)):

In the manuscript, Jia et al. report MakA as a new pH-dependent cholesterol-binding pore-forming toxin. Interaction between MakA and cholesterol were confirmed by a few methods. The binding was obvious below pH 6.8. MaKA is structurally similar to ClyA, but the binding with cholesterol is pH dependent. Residues I236 and I237 were found be critical for interaction with cholesterol, different from cholesterol-dependent cytolysins. MakA forms pores on membranes with cholesterol at pH 5.4 but not pH 6.8, indicating complicated regulation and multiple states of MakA. MakA in host cells will form pores at low pH and cause membrane damage that recruits ESCRT machinery and non-canonical autophagy. These findings reveal the molecular mechanisms of MakA regulation of host signaling pathways. The manuscript is well prepared but the following issues need to be addressed:

Major points:

1. Introduction, Paragraph 5: "... and subsequent inhibition of canonical and non-canonical autophagic responses.", "inhibition" is causing confusion and needs to be explained as MakA is inducing non-canonical autophagy throughout the manuscript.

Re1: Our previous study (Corkery et al. JCS 2021, jcs252015) demonstrated that the MakA-induced endolysosomal membrane aggregate leads to the inhibition of subsequent canonical autophagy and autophagy-related processes including the unconventional secretion of IL-1 β (likely a result of sequestration of the autophagy machinery at the aggregate). We have adjusted the wording in the introduction to avoid any confusion (line 86-87)

2. Fig. 3A: Did the cryo-EM data result in structure models (high-res or not)? This needs to be discussed and explained. The structure information will be helpful and strengthen the manuscript. Please estimate the number of monomers in a pore. How many monomers are taken up in an endosome? Will the apparent concentrations be high enough? μ M?

Re2: Unfortunately, after extensive attempts at 3D classification and reconstruction, we had to conclude that the molecules within the pore are too flexible to render a good quality 3D map. So, we could not get the exact number of monomers in a pore. However, the 2D propeller top view of the pore (Fig. 3A) is similar in dimension and organization to the top view of the recently described tubular structure (Nadeem et al. eLife 2022, e73439). The discussion of how our data relates to the tubular structure of MakA has been significantly expanded (line 319-342, see also Re4).

With current results and approaches, we could not estimate exactly how many monomers are taken up in an endosome. But, we have shown previously that the effect of MakA in cells is dose-dependent (minimal 250 nM) (Fig. 3A,B, Corkery et al. JCS 2021, jcs252015). During endocytosis the

effective concentrations of MakA would increase in endosomes. Moreover, binding to cell-surface receptors would also increase the effective concentrations of MakA. These results correlate with the MakA pore-formation in a dose-dependent manner (0.625-20 μ M) (Fig. 3B).

3. Low pH triggers MakA-induced membrane damage and non-canonical autophagy, Paragraph 3: "Treatment with both BafA1 and MakA results in LC3 II accumulation similar to BafA1 alone, suggesting inhibition of MakA-dependent LC3 lipidation." This is only one possibility though the text below shows that it may be the situation. Other possibilities are inhibition of BafA1-induced lipidation or LC3II is consumed out. This sentence needs to be revised to fix the logical problem.

Re3: Thank you for pointing out this logical problem. We have adjusted the text to emphasize that BafA1-mediated inhibition of MakA-induced LC3 lipidation is just one of several hypotheses that would explain the non-additive effect of BafA1 + MakA treatment (line 231-234).

4. Discussion, Last Paragraph: "tubular structure of MakA at pH 6.5" should be discussed in detail as it is related to the findings described in the manuscript. Did the authors observe any similar structures? Any indication for function or regulation? Top views of 2D averages seem to be similar with tubular structure top view, do the authors have any idea about the aggregate structure? Could it be tubular?

Re4: We did not observe tubular structures at pH 5.4 under these conditions (Fig. 3A). MakA oligomer in solution (MakA_O) showed star-shaped structures, but no tube formation (Fig. R1). It is not yet clear if the MakA tubes form only under specific conditions *in vitro* or if they have certain physiological functions. However, the similar propeller-shape of 2D top views of the pore and the tube suggests that the tubular structures may somewhat reflect the organization of the pore and the conformational changes during pore formation.

The discussion of how our data relates to the recently described tubular structure of MakA has been significantly expanded (line 319-342).

Fig R1. Representative negative-stain micrograph of MakA_O in solution.

5. The similarity between MakA and ClyA should be explained with more details to illustrate possible mechanisms for cholesterol-binding and pH regulation. A structural comparison with cholesterol-dependent cytolysins will also provide important information.

Re5: The discussion surrounding MakA, cholesterol-dependent cytolysins, other pH-dependent PFTs and ClyA has been expanded to illustrate possible mechanisms for cholesterol-binding and pH regulation (line 283-291; line 300-342).

“Cholesterol interaction is the hallmark of the CDC family with a conserved cholesterol-binding motif (Thr-Leu pair) identified (Farrand et al., 2010). A plausible binding mechanism of CDCs has been proposed whereby the leucine inserts into the lipid bilayer and interacts with hydrophobic rings of cholesterol. The threonine then forms a hydrogen bond with the 3-hydroxyl group of cholesterol (Hotze and Tweten, 2012). Unlike the traditional CDC cholesterol-binding motif, we identified an Ile-Ile pair (I236 and I237) in MakA essential for its interaction with membrane cholesterol. The adjacent S235 is dispensable for this interaction, yet the 3-hydroxyl of cholesterol appears to be somewhat important. This suggests that MakA features a novel cholesterol-binding mechanism different from other CDCs and in keeping with the notion that MakA displays low structure similarity with CDCs.”

“In earlier studies, several PFTs have been identified showing acidic pH-induced membrane insertion, such as anthrax toxin from *Bacillus anthracis* (Jiang et al., 2015), colicin A from *E.coli* (Davidson et al., 1985), diphtheria toxin (DT) from *Corynebacterium diphtheria* (Rodnin et al., 2016), LLO (Schuerch et al., 2005), vacuolating cytotoxin A (VacA) from *Helicobacter pylori* (de Bernard et al., 1995), perfringolysin (PFO) from *Clostridium perfringens* (Nelson et al., 2008), lysenin from earthworm *Eisenia foetida* (Munguira et al., 2017), Tc toxins (Meusch et al., 2014) and a recently reported mammalian pore-forming protein perforin-2 (PFN2) (Ni et al., 2020). Most of these PFTs undergo a pre-pore to pore transition under low pH with plausible pH-sensitive domains reported. The protonation of three glutamic acids in a short α -helix of lysenin were identified to be involved in low pH-induced membrane insertion (Munguira et al., 2017). For anthrax toxin, it was demonstrated that one $2\beta_{10}$ - $2\beta_{11}$ loop (N422, D425) served as a pH sensor to trigger conformation changes for pre-pore to pore conversion (Jiang et al., 2015). The putative pH-sensing mechanism in MakA is still unclear, but likely relates to protonation of specific amino acids under low pH. In addition to the closed pore formation of PFTs, oligomerization occasionally ends up with an incomplete pore-like arc-shape. By using real-time atomic force microscopy (AFM), Cryo-EM and atomic structure fitting, it was demonstrated that suilysin assembly resulted in not only ring-shaped pores but also kinetically trapped arc-shaped oligomers, both capable of perforating the membrane (Leung et al., 2014). Arc-shaped structures of LLO were also confirmed by using high-speed AFM, which revealed arc pores inserted into membrane causing damage (Ruan et al., 2016). Membrane attack complex (MAC), an immune system PFT, was also shown to form pores in a closed and open conformation (Menny et al., 2018).

The crystal structure of MakA monomer is similar to toxins in the ClyA α -PFT family (Churchill-Angus et al., 2021; Dongre et al., 2018; Ganash et al., 2013; Herrera et al., 2022; Madegowda et al., 2008; Mueller et al., 2009; Wilson et al., 2019). Recently, a tubular structure of MakA oligomer at pH 6.5 showed that MakA undergoes conformational changes from a soluble to a membrane-bound state (Nadeem et al., 2022). The propeller-shape of the 2D averages of MakA (Fig. 3A) is similar in dimension and organization to the top view of the tubular structure. The Ile-Ile pair (I236 and I237) essential for cholesterol interaction is located within one of the lipid interaction regions (222-242) in the tubular structure (Nadeem et al., 2022) (Fig. 7B). It is not yet clear if the MakA tubes form only under specific conditions *in vitro* or if they have certain physiological functions. However, the similar propeller-shape of 2D top views of the pore and the tube suggests that the tubular structures may somewhat reflect the organization of the pore and the conformational changes during pore formation.

Earlier studies have shown that other PFTs from the ClyA family (ClyA, HBL-B and Nhe) also require cholesterol for their insertion into membranes (Dal Peraro and van der Goot, 2016). Cholesterol stimulates ClyA pore formation by a dual-mode of selectively stabilizing a protomer-like conformation and bridging interactions in the protomer-protomer interfaces. Tyr27 in the N-terminal helix of ClyA was shown to be a key determinant for ClyA-cholesterol interaction, whereas the cholesterol-interacting Ile-Ile pair of MakA is located within the β -tongue in the monomeric state (Dongre et al., 2018; Nadeem et al., 2022; Sathyanarayana et al., 2018). Upon membrane-binding,

the N-terminal helix of ClyA folds out to insert into the membrane, where cholesterol stabilizes the conformation and the protomer-protomer interactions (Fig. 7A). In contrast, according to the tubular structure model of MakA, the β -tongue together with $\alpha 4$ and $\alpha 5$ detaches from the core of the protein and forms two extended helices (Fig. 7B). It remains elusive whether cholesterol plays a similar role in MakA pore formation as it does in ClyA pore formation. Nevertheless, the pH-dependent pore-forming activity of MakA appears to be unique amongst the ClyA α -PFTs.”

Fig 7. Comparison between ClyA and MakA structures. (A) Crystal structures of ClyA monomer (PDB 1QOY) (Walace et al., 2000) (left) and single protomer from the ClyA pore (PDB 2WCD) (Mueller et al., 2009) (right). The membrane interacting domains, β -tongue (yellow), the N-terminal helix (α A1, orange) and the key cholesterol-binding residue Y27 (red), are highlighted. (B) Structures of MakA monomer (PDB 6EZV) (Dongre et al., 2018) and a subunit of the MakA tube (PDB 7P3R) (Nadeem et al., 2022). The regions that undergo conformational changes are highlighted: β -tongue (yellow), $\alpha 4$ (blue) and $\alpha 5$ (light blue). The cholesterol-interacting Ile-Ile pair is shown in red.

Minor points:

1. Introduction, Paragraph 3: "... in response to activation of several non-autophagic pathways (11) ...", "non-autophagic" should be "non-canonical".

Re6: "non-autophagic" has been changed to "non-degradative pathways" (line 74) to remain consistent with the nomenclature used in the cited literature.

2. Fig. S3E: "S235S" should be S235A.

Re7: This has been corrected

3. For clarity change

Importantly, co-treatment with MakA and BafA1 completely blocks MakA-induced LC3 lipidation (Fig.

5A, 5B).

To

Importantly, co-treatment with MakA and BafA1 in cells deficient for basal autophagy completely blocks MakA-induced LC3 lipidation (Fig. 5A, 5B).

R8: This has been changed (line 238)

Reviewer #2 (Comments to the Authors (Required)):

This paper finds that the Pore forming toxin MakA interacts with Cholesterol and forms oligomers at low pH. Another major finding is a mutant that locks the toxin in an oligomeric state. The results are of interest and generally well conducted. The work expands and work by others with interesting new findings. The work itself is solid. The primary issues with the paper are the overinterpretation of some results and some conclusions not fully supported and lack of consideration of results in context of published literature. A redrafted manuscript (especially discussion) should address all of the following major and minor comments.

Major comments

1) The paper discusses multiple cholesterol dependent cytolysins (CDC) without mentioning the pivotal work by Michael Jennings in 2014 (PNAS <https://doi.org/10.1073/pnas.1412703111>) and 2020 (Science Advances DOI: 10.1126/sciadv.aaz4926). These studies reveal CDCs use glycans as cellular receptors which drives their cell tropism, in contrast to prior considerations that they bind cholesterol directly in membranes. This is important here as a recent 2022 paper by Herrera et al showed that while MakA is cytotoxic at high concentrations on many different cells, it shows cell tropism similar to many of the CDCs, suggesting it does use a receptor. Further, it was functional at nanomolar concentrations when combined with MakB and MakE, consistent with high affinity engagement. The finding that MakA CAN bind cholesterol and may even depend on cholesterol and pH for insertion and/or oligomerization are well done and move the field forward in our understanding of MakA. However, the data in this paper do not support that MakA is a receptor. Most importantly, the binding affinity is in the micromolar ranges and MakE and MakB were found not to bind cholesterol at all. It is suggested the authors review this body of literature and update the manuscript to accurately reflect the role of cholesterol in CDCs and remove all comments that cholesterol may be a receptor for MakA. The discussion should be tempered that cholesterol is "important" for MakA interactions and possibly insertion/oligomerization when functioning as a single toxin, which may be different that the role of cholesterol when MakA is part of the high affinity tripartite toxin that has specific cell tropism.

Re1: We thank the reviewer for the constructive comments. The discussion has been redrafted (line 283-299) to provide a more detailed perspective on the role of cholesterol in CDCs, and how this might relate to our findings on MakA.

Our observation that MakA^{I236D&I237D} (cholesterol-binding deficient) showed no sign of plasma membrane binding in MEF cells (Fig. 2E) and, consequently, were unable to induce cholesterol-rich aggregate formation (Fig. 2E-F) or LC3 lipidation (Fig. 2G) suggests that the cholesterol-MakA interaction is important (if not essential) for plasma membrane binding. However, we acknowledge that we cannot rule out co-operation from another receptor and have adjusted the text (line 290-298) and model (Fig. 6) to reflect this.

2) Although there is no overlap in named authors, scientists from the same institution previously

reported that this toxin binds to phosphatitic acid. Whereas Fig 1B shows no interaction. Discussion should address the difference in the two studies and support why this study should take precedence? Perhaps technical?

Re2: Nadeem et al. observed binding of MakA to PC:PA liposomes by the liposome sedimentation assay (Nadeem et al. PLoS Pathog. 2021, 17(3):e1009414). We repeated the experiment of PA binding under the same conditions as those used in the Nadeem paper, i.e. in the same buffer pH 7 and liposomes containing 50% PA. We used 5 μ M MakA which is 19-fold above the reported K_d value in the Nadeem paper to saturate the binding. However, no binding to PA was observed under these conditions (Fig. R2). Interestingly, under the same conditions, we found that His-tagged MakA binds to liposomes with 50% PA, similar to the liposome sedimentation assay presented in the Nadeem paper (Fig. 1G in Nadeem paper). Therefore, the observed MakA-PA binding could be an artifact of His-tag. We have communicated with the authors about this matter.

Fig R2. SDS-PAGE and analysis of liposome sedimentation assays of MakA's or His-MakA's interaction with PA. 5 μ M of MakA or His-MakA was incubated with DOPC:DOPA (1:1) liposomes (5 mM) for 1 h at 37 $^{\circ}$ C in HEPES buffer (10 mM HEPES, 150 mM NaCl, pH 7.0). Liposomes were extruded through 0.1 μ m polycarbonate membrane 11 times by Avanti Mini Extruder. After incubation, the protein-liposome mixtures were centrifuged for 1 h at 4 $^{\circ}$ C with 12700 rpm to separate supernatant and pellet. Data were averaged from three independent replicates. S: supernatant, P: Pellet.

3) As part of point 1, Fig. 1 and S1 nicely show that MakA interactions with cholesterol loaded liposomes by two methods. However, it is important they note that, cholesterol CAN interact with MakA in vitro at a pH ideal for MakA-induced LC3 lipidation process. This does not necessarily lead to their conclusion that MakA will bind to membrane cholesterol in cells. Further, when the toxin would be in this environment is not clear? The model seems to suggest interaction prior to endocytosis with assembly occurring at pH6.5 in the acidified vacuole. This does not support a role of cholesterol as a receptor for MakA, as this should occur at the plasma membrane surface, well before acid conditions encountered. The model should be amended to accurately consider when and where MakA would encounter pH and cholesterol as imaged in Fig 6 at the plasma membrane surface is not likely the location

Re3: As mentioned above (pt.1 response), we are unable to rule out the involvement of an additional receptor for MakA and have adjusted the text (line 291-299) and model (Fig. 6) to reflect this. The question of when and where MakA might encounter these conditions raises an interesting point when considered from the perspective of host-infection. Normal serum pH (7.35-7.45) is outside of the range which promotes MakA-cholesterol interaction, but the small intestine provides a pH gradient (6.15-7.88) extending into the interaction range. In this sense, there could be a region within

the small intestine where MakA encounters cholesterol at the plasma membrane at the correct pH to promote its direct interaction. This, of course, is highly speculative and would require *in vivo* studies of GI infection to confirm, but is interesting food-for-thought.

4) The results section regarding Fig 5 are very complicated and the text is difficult to follow making it hard to assess the results regarding non-canonical autophagy although the data seem to support the conclusions drawn.

Re4: We have adjusted the text surrounding Figure 5 (line 222-252) in an attempt to clarify the intention and interpretation of experiments performed using the panel of autophagy deficient cell lines.

5) The discussion is rather abrupt and does not integrate well with the literature in this field and the model as presented in Figure 6 is not well supported.

Re5: The discussion has been redrafted to better integrate with the existing literature surrounding PFTs.

Figure 6 has been adjusted to include the possibility of an unidentified receptor regulating plasma membrane binding.

Minor points

1) *Vibrio cholerae* and *V. cholerae* are spelled wrong throughout the paper.

-this has been corrected throughout.

2) Figure 1E Seems to present only one image. Are these cells fixed? is the toxin on the surface or inside the cell? Cannot tell from these images. Quantification shows only a single cell.

-Figure 1E shows cells (Cos7/HeLa) treated with A568 labelled MakA then fixed and stained for cholesterol (filipin). As described previously (Corkery et al. JCS 2021), the toxin is observed at both the plasma membrane, and accumulated at the internalized endolysosomal membrane aggregate. Line scans were performed on internalized MakA-induced aggregates to show colocalization between the toxin and cholesterol inside the cell. Additional line scans have been added to the Figure and a reproducibility statement added to the figure legend for clarification.

3) Please indicate in legend that representative images are representative of How many (n) cells and experiments?? Reproducibility information on microscopy is missing from the legends.

-reproducibility information has been added where appropriate

4) Crystal structures are all now published. Please cite all references when using the PDB ID#. See papers by Herrera et al Infect Immun 2022 and by Nadeem et al PNAS 2021.

-citations have been added where appropriate.

5) Papers by others support other models of death than non-canonical autophagy. How do results here compare with recent papers on MakA? How would the model account for evidence this works at much lower concentration in concert with MakB and MakE in specific cell lines

-we expect the induction of autophagy is protective against cell death, not the cause. This is consistent with published data showing autophagy deficient cells (ATG5 -/-) display increased sensitivity to MakA (Nadeem et al. PLOS Path. 2021). How/when MakA works alone vs. as a

component of the tripartite cytolytic complex remains an interesting question which would require a better understanding of the relative expression levels of the Mak proteins at different stages of disease progression to begin answering.

Reviewer #3 (Comments to the Authors (Required)):

Summary

The authors characterize binding of a pore forming toxin MakA to cholesterol. They identify key residues for binding, and show that binding requires an acidic pH. Using CryoEM they show that MakA can form 3nm pores, in at least the conditions they used. In prior studies they showed that MakA induces formation of endosomal aggregates positive for autophagic markers (e.g. LC3). Here they show that MakA induces these cellular aggregates through generating small pores, and independently of the autophagic response. Overall, the experiments are well designed and the data are of high quality. The paper is of interest to microbiologists since pore forming toxins are major bacterial virulence factors, but also to cell biologists since it examines how endosomes/lysosomes can be repaired. The main weakness of the paper is they do not take advantage of prior structural data available for MakA.

Major comments

-An important consideration is the nature of MakA protein monomers. How stable is MakA protein at 37 degrees at different pH, independently of any lipids? Does it lose membrane binding/perforating activity due to loss of stability of the protein at certain pH?

Re1: As shown in Fig. 1C, the control experiments of MakA in buffer at different pHs showed that MakA is stable in solution (no aggregate and oligomerization).

-The cryo EM images in Figure 3A are impressive. Can a combined model be generated? Also, is it expected that these maximally 3nm pores, or are these images representing an intermediate in the assembly of larger pores? How do these images compare to other PFTs?

Re2: Unfortunately, after extensive attempts at 3D classification and reconstruction, we had to conclude that the sample is too flexible to render a good quality 3D map. The discussion surrounding the comparison of MakA pores with those of other PFTs has been expanded (line 277-342).

-pg8, last paragraph of Discussion. The authors acknowledge that MakA was shown in another paper (ref 49) to oligomerize into tubular structures. That paper has important structural data that should be leveraged as much as possible to provide insight into the data acquired here. Importantly, are the authors visualizing an alternate form of the toxin? Or do they think the big tube-like oligomers are forming within endosomes after acidification?

Re3: The discussion of how our data relates to the recently described tubular structure of MakA and compares to the ClyA structures has been significantly expanded (line 319-342). We did not observe tubular structures at pH 5.4 under these conditions (Fig. 3A). It is not yet clear if the MakA tubes form only under specific conditions *in vitro* or if they have certain physiological functions. However, the similar propeller-shape of 2D top views of the pore and the tube suggests that the tubular structures may somewhat reflect the organization of the pore and the conformational changes during pore formation.

“The crystal structure of MakA monomer is similar to toxins in the ClyA α -PFT family (Churchill-Angus et al., 2021; Dongre et al., 2018; Ganash et al., 2013; Herrera et al., 2022; Madegowda et al., 2008; Mueller et al., 2009; Wilson et al., 2019). Recently, a tubular structure of MakA oligomer at pH 6.5

showed that MakA undergoes conformational changes from a soluble to a membrane-bound state (Nadeem et al., 2022). The propeller-shape of the 2D averages of MakA (Fig. 3A) is similar in dimension and organization to the top view of the tubular structure. The Ile-Ile pair (I236 and I237) essential for cholesterol interaction is located within one of the lipid interaction regions (222-242) in the tubular structure (Nadeem et al., 2022) (Fig. 7B). It is not yet clear if the MakA tubes form only under specific conditions *in vitro* or if they have certain physiological functions. However, the similar propeller-shape of 2D top views of the pore and the tube suggests that the tubular structures may somewhat reflect the organization of the pore and the conformational changes during pore formation.

Earlier studies have shown that other PFTs from the ClyA family (ClyA, HBL-B and Nhe) also require cholesterol for their insertion into membranes (Dal Peraro and van der Goot, 2016). Cholesterol stimulates ClyA pore formation by a dual-mode of selectively stabilizing a protomer-like conformation and bridging interactions in the protomer-protomer interfaces. Tyr27 in the N-terminal helix of ClyA was shown to be a key determinant for ClyA-cholesterol interaction, whereas the cholesterol-interacting Ile-Ile pair of MakA is located within the β -tongue in the monomeric state (Dongre et al., 2018; Nadeem et al., 2022; Sathyanarayana et al., 2018). Upon membrane-binding, the N-terminal helix of ClyA folds out to insert into the membrane, where cholesterol stabilizes the conformation and the protomer-protomer interactions (Fig. 7A). In contrast, according to the tubular structure model of MakA, the β -tongue together with $\alpha 4$ and $\alpha 5$ detaches from the core of the protein and forms two extended helices (Fig. 7B). It remains elusive whether cholesterol plays a similar role in MakA pore formation as it does in ClyA pore formation. Nevertheless, the pH-dependent pore-forming activity of MakA appears to be unique amongst the ClyA α -PFTs.”

Fig 7. Comparison between ClyA and MakA structures. (A) Crystal structures of ClyA monomer (PDB 1QOY) (Wallace et al., 2000) (left) and single protomer from the ClyA pore (PDB 2WCD) (Mueller et al., 2009) (right). The membrane interacting domains, β -tongue (yellow), the N-terminal helix ($\alpha A1$, orange) and the key cholesterol-binding residue Y27 (red), are highlighted. (B) Structures of MakA monomer (PDB 6EZV) (Dongre et al., 2018) and a subunit of the MakA tube (PDB 7P3R) (Nadeem et al., 2022). The regions that undergo conformational changes are highlighted: β -tongue (yellow), $\alpha 4$ (blue) and $\alpha 5$ (light blue). The cholesterol-interacting Ile-Ile pair is shown in red.

Minor comments:

-page and line numbers would help this reviewer

-this has been added

-Should MakA be classified as a pore-forming toxin or a cholesterol-dependent cytolysin? What is the consensus for this?

-our data shows that MakA is a novel cholesterol-binding, pH-dependent pore-forming toxin (PFT). MakA has unique features, a distinct cholesterol-binding motif and displays pH-dependent pore formation, which are mechanistically different from currently identified ClyA family PFTs and cholesterol-dependent cytolysins (CDCs) (see discussions line 283-342).

-pg2, second paragraph "pores range from a few nm to 50 nm". This is not entirely accurate... monomers can affect membranes with channel like activity, and form other structures such as arcs, but they are not really pores at that point (in the way that we typically think about them). Some of the atomic force microscopy studies of other toxins should be cited and discussed here.

-thanks for this clarification. The introduction (line 58-60) and discussion (line 312-318) have been adjusted to include these arcs-shaped structures.

-pg3, please define "PM mix liposomes"?

-the PM mix liposomes are defined in the text (line 112 – "PM mix liposomes containing PC/PE/PS/PI/cholesterol/sphingomyelin (5:1:0.5:0.5:2:1)")

-pg7, "In contrast, MakA pore on the membrane (~ 3 nm) is smaller, which is doable for the passage of ions or small molecules but not proteins". I'm not sure if this is true, since the type 3 secretion system from bacteria has a narrow shaft and yet translocates proteins into host cells. Also, the word "doable" is a slang term.

-this has been rephrased (line 278-279) to clarify that the pore is too small for most folded proteins to easily pass through

September 13, 2022

RE: JCB Manuscript #202206040R

Prof. Yaowen Wu
Umeå University
Department of Chemistry
Linnaeus väg 6
Umeå 90187
Sweden

Dear Prof. Wu:

Thank you for submitting your revised manuscript entitled "V. Cholerae MakA is a cholesterol-binding pore-forming toxin that induces non-canonical autophagy". We would be happy to publish your paper in JCB pending final revisions necessary to meet our formatting guidelines (see details below).

A. MANUSCRIPT ORGANIZATION AND FORMATTING:

Full guidelines are available on our Instructions for Authors page, <https://jcb.rupress.org/submission-guidelines#revised>. Submission of a paper that does not conform to JCB guidelines will delay the acceptance of your manuscript.

1) Text limits: Character count for Articles is < 40,000, not including spaces. Count includes abstract, introduction, results, discussion, and acknowledgments. Count does not include title page, figure legends, materials and methods, references, tables, or supplemental legends.

2) Figures limits: Articles may have up to 10 main text figures.

3) Figure formatting: Scale bars must be present on all microscopy images, including inset magnifications. Molecular weight or nucleic acid size markers must be included on all gel electrophoresis images.

**Please add scale bars to images in Figure 1D and insets in Supplementary Figure 5A

**Please add molecular weight markers to all gel images.

4) Statistical analysis: Error bars on graphic representations of numerical data must be clearly described in the figure legend. The number of independent data points (n) represented in a graph must be indicated in the legend. Statistical methods should be explained in full in the materials and methods. For figures presenting pooled data the statistical measure should be defined in the figure legends. Please also be sure to indicate the statistical tests used in each of your experiments (either in the figure legend itself or in a separate methods section) as well as the parameters of the test (for example, if you ran a t-test, please indicate if it was one- or two-sided, etc.). Also, if you used parametric tests, please indicate if the data distribution was tested for normality (and if so, how). If not, you must state something to the effect that "Data distribution was assumed to be normal but this was not formally tested."

5) Abstract and title: The abstract should be no longer than 160 words and should communicate the significance of the paper for a general audience. The title should be less than 100 characters including spaces. Make the title concise but accessible to a general readership.

6) Materials and methods: Should be comprehensive and not simply reference a previous publication for details on how an experiment was performed. Please provide full descriptions in the text for readers who may not have access to referenced manuscripts.

7) Please be sure to provide the sequences for all of your primers/oligos and RNAi constructs in the materials and methods. You must also indicate in the methods the source, species, and catalog numbers (where appropriate) for all of your antibodies. Please also indicate the acquisition and quantification methods for immunoblotting/western blots.

8) Microscope image acquisition: The following information must be provided about the acquisition and processing of images:

- a. Make and model of microscope
- b. Type, magnification, and numerical aperture of the objective lenses
- c. Temperature
- d. Imaging medium

- e. Fluorochromes
- f. Camera make and model
- g. Acquisition software
- h. Any software used for image processing subsequent to data acquisition. Please include details and types of operations involved (e.g., type of deconvolution, 3D reconstitutions, surface or volume rendering, gamma adjustments, etc.).

10) Supplemental materials: There are strict limits on the allowable amount of supplemental data. Articles may have up to 5 supplemental figures. Please also note that tables, like figures, should be provided as individual, editable files. A summary of all supplemental material should appear at the end of the Materials and methods section.

13) ORCID IDs: ORCID IDs are unique identifiers allowing researchers to create a record of their various scholarly contributions in a single place. At resubmission of your final files, please consider providing an ORCID ID for as many contributing authors as possible.

Please note that JCB now requires authors to submit Source Data used to generate figures containing gels and Western blots with all revised manuscripts. This Source Data consists of fully uncropped and unprocessed images for each gel/blot displayed in the main and supplemental figures. Since your paper includes cropped gel and/or blot images, please be sure to provide one Source Data file for each figure that contains gels and/or blots along with your revised manuscript files. File names for Source Data figures should be alphanumeric without any spaces or special characters (i.e., SourceDataF#, where F# refers to the associated main figure number or SourceDataFS# for those associated with Supplementary figures). The lanes of the gels/blots should be labeled as they are in the associated figure, the place where cropping was applied should be marked (with a box), and molecular weight/size standards should be labeled wherever possible.

B. FINAL FILES:

**The license to publish form must be signed before your manuscript can be sent to production. A link to the electronic license to publish form will be sent to the corresponding author only. Please take a moment to check your funder requirements before

choosing the appropriate license.**

Thank you for this interesting contribution, we look forward to publishing your paper in Journal of Cell Biology.

Sincerely,

Craig Roy
Monitoring Editor
Journal of Cell Biology

Tim Fessenden
Scientific Editor
Journal of Cell Biology

Reviewer #1 (Comments to the Authors (Required)):

The authors have satisfied this reviewers concerns.

Reviewer #2 (Comments to the Authors (Required)):

The authors have responded well to my prior recommendation of updating introduction and discussion to integrate current literature and to remove comments regarding cholesterol serving as a receptor. They have modified the model figure 6 to match better their finding in light of the literature.

I have no further comments

Reviewer #3 (Comments to the Authors (Required)):

The authors have addressed my comments.